# Automated Composition of Agents: A Knapsack Approach for Agentic Component Selection

**Michelle Yuan**[*][†] , **Khushbu Pahwa**[*] , **Shuaichen Chang**
**Mustafa Kaba**, **Jiarong Jiang**, **Xiaofei Ma**, **Yi Zhang**, **Monica Sunkara**
AWS Agentic AI
`michelle.yuan@oracle.com`
`{pahwakhu,cshuaich,mdkaba,jiarongj,xiaofeim,yizhngn,sunkaral}@amazon.com`

## Abstract

Designing effective agentic systems requires the seamless composition and integration of agents, tools, and models within dynamic and uncertain environments. Most existing methods rely on static, semantic retrieval approaches for tool or agent discovery. However, effective reuse and composition of existing components remain challenging due to incomplete capability descriptions and the limitations of retrieval methods. Component selection suffers because the decisions are not based on capability, cost, and real-time utility. To address these challenges, we introduce a structured, automated framework for agentic system composition that is inspired by the knapsack problem. Our framework enables a composer agent to systematically identify, select, and assemble an optimal set of agentic components by jointly considering performance, budget constraints, and compatibility. By dynamically testing candidate components and modeling their utility in real-time, our approach streamlines the assembly of agentic systems and facilitates scalable reuse of resources. Empirical evaluation with Claude 3.5 Sonnet across five benchmarking datasets shows that our online-knapsack-based composer consistently lies on the Pareto frontier, achieving higher success rates at significantly lower component costs compared to our baselines. In the single-agent setup, the online knapsack composer shows a success rate improvement of up to 31.6% in comparison to the retrieval baselines. In multi-agent systems, the online knapsack composer increases success rate from 37% to 87% when agents are selected from an agent inventory of 100+ agents. The substantial performance gap confirms the robust adaptability of our method across diverse domains and budget constraints.

## 1 Introduction

The design of effective agentic systems sits at the frontier of artificial intelligence research, promising autonomous agents capable of sophisticated reasoning, tool manipulation, and collaborative problem-solving. Yet as the ecosystem of AI components expands, with proliferating models, APIs, and specialized agents, a critical bottleneck emerges: the paradox of choice [23]. While modular reuse offers clear advantages over building systems from scratch, developers face a combinatorial explosion of possible configurations, each with hidden constraints and unpredictable interactions. Traditional approaches relying on manual curation or metadata-based retrieval [29] struggle with three fundamental limitations: opaque capability descriptions that rarely match real-world performance, myopic selection criteria that ignore cost-utility trade-offs, and static architectures that break down when requirements evolve.

---

[*]These authors contributed equally.
[†]Work completed while at Amazon. Currently at Oracle AI.

39th Conference on Neural Information Processing Systems (NeurIPS 2025).

In this paper, we introduce agent composition as a knapsack problem, where an AI system designer must select components that deliver optimal performance under cost constraints. We present the composer agent, which is responsible for selecting components through an iterative process of discovery and adaptation. Rather than treating existing tools or agents as fixed building blocks, the composer continuously probes their actual capabilities through targeted testing. The sandbox trials measure not just what a component claims to do, but how reliably it performs under varying conditions and interactions. For single-agent systems, this means assembling the right tools for the task while balancing accuracy against computational expense; for multi-agent teams, it involves orchestrating subagents whose skills complement rather than duplicate one another.

What distinguishes our approach is estimating values of agentic components based on real-time performance. Where prior work treated component selection as a one-time decision based on static metadata, the composer refines its choices through empirical validation. When composing an information-seeking agent, for instance, it might initially select a specialized search tool for scientific queries, only to discover through testing that a single, generalized search tool can accommodate queries across various domains. On the other hand, a task might require deep expertise in medical fields, so the scientific search tool would be more useful than a generalized web search. This dynamic estimation capability proves particularly valuable in real-world deployments where requirements and components shift unpredictably. Our work makes three contributions:

1. A formalization of agent composition as a constrained optimization problem that jointly considers capability, cost, and compatibility, bridging gaps between modular AI design [10] and operations research [41].

2. A workflow for the composer agent that parses task descriptions into skills, assesses utility values of agentic components based on real-time testing, and then returns optimal agentic components for a particular domain.

3. Empirical validation across diverse domains showing consistent improvements in cost-adjusted performance (up to 80% performance gains over retrieval-based baselines).

Design of machine learning systems is critical as poor organization leads to unstable dependencies, pipeline jungles, and many other types of hidden technical debt [28]. Our work is a step towards optimizing this process to reduce these issues and maintain a well-composed agentic system. As we explore in subsequent sections, our approach is applicable to composing both single agents and multi-agent systems.

## 2   Related Work

**Tool Retrieval**   The novelty of AI agents stems from the integration of APIs with LLMs [27, 24]. Since tools are a critical component of agents, prior works have focused on retrieval approaches to select the most appropriate tools. Qin et al. [25] collect a large dataset of tools based on RapidAPIs and train a BERT-based tool retriever. Shi et al. [29] note that tool retrieval is a difficult problem as commonly used retrievers fail to capture user intent and select the most relevant tools. RAG-MCP [9] combines RAG with the Model Context Protocol (MCP) to improve how LLMs select external tools. Recently, more work has emphasized the vulnerability of agents due to poor selection of tools [6, 22].

**Agentic System Design and Optimization**   Hu et al. [10] introduce the problem of automated design of agentic systems (ADAS), where the goal is to "automatically create powerful agentic system designs, including inventing novel building blocks and/or combining them in new ways". Conceptually, the building blocks of ADAS includes novel prompts, tool use, and workflows. In our paper, we focus on a subset of ADAS, which is automated composition of agentic systems. Our setting assumes that the underlying building blocks exist and the algorithmic challenge is to select the most optimal subset of those components. Common solutions to this problem include tool retrieval and agent selection. For agent selection, existing works treat the problem as graph optimization where nodes are the individual agents and edges represent communication channels. DyLAN [17] introduce the agent selection problem and train a feed-forward network to optimize the selection. AgentPrune [40] extend selection to target improving communication redundancy. Multi-agent Architecture Search [39] optimizes an agentic supernet and then will sample a multi-agent architecture in a query-dependent manner. Wu et al. [35] jointly optimize agent prompt and tool descriptions to improve workflow efficiency. The component selection problem also relates to

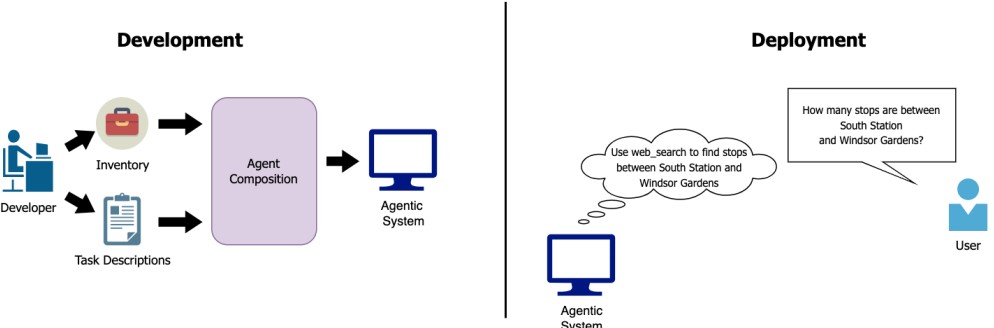

Figure 1: The problem of agent composition assumes there is existing inventory of agentic components and a task description. Using these inputs, the agent composition solution should select the most relevant components to build an agent (left). The agentic system can then be deployed as it has the correct set of components to accomplish goals as set by the user at inference.

Distributed Constraint Optimization Problems (DCOP) [8], where traditional agents assign values to variables to minimize a global cost function subject to constraints. DCOP techniques have been applied to service selection and composition problems, particularly for QoS-aware web service composition [4].

**Knapsack Algorithms**   The knapsack problem, a classic optimization challenge, has been extensively studied in algorithmic research [19, 7, 14]. Offline algorithms, such as dynamic programming [2] and branch-and-bound methods [18], assume complete knowledge of all items in advance, enabling optimal solutions for static inputs. In contrast, online knapsack algorithms process items sequentially without future information. A commonly known approach is the ZCL algorithm [41] which proposes a threshold for the value-to-weight ratio and is dynamically configured based on the capacity of the knapsack. Other theoretically competitive approaches further extend the ZCL algorithm [11, 16].

## 3   Problem Definition

Hu et al. [10] introduce the problem of automated design of agentic systems (ADAS), where the goal is to automatically build and configure the components of an agentic system. Rather than manually engineering agent architectures, ADAS aims to automate the entire procedure through algorithmic discovery. Components of agents include tools, models, prompts, and workflows. Our work addresses a critical subproblem within this vision: automated agent composition, which focuses not on inventing new building blocks from scratch, but on optimally selecting and combining existing components from inventories to meet specific task requirements.

We define the problem of **agent composition** as follows (Figure 1). Assume that there is a target task $\tau$ with a description $x$ and a budget $B$ for the agentic system. Assume that there exists a set $\mathcal{A}$ of components $a_i$ where each component has a cost $c_i$ and a description $d_i$. Let $p_\tau(\mathcal{S})$ denote the probability of **success rate** for an agentic system built on a subset $\mathcal{S}$ for task $\tau$. The success probability should reflect both the performance of the individual components and their combined effectiveness when operating together. The goal of agent composition is to find the optimal subset $\mathcal{S}^*$ of components for this domain:

$$\mathcal{S}^* = \arg\max_{\mathcal{S} \subseteq \mathcal{A}} p_\tau(\mathcal{S}) \quad \text{subject to} \quad \sum_{a_i \in \mathcal{S}} c_i \leq B \tag{1}$$

This formulation directly mirrors the **classical knapsack problem**, where components correspond to items with associated weights (costs) and values (success rates). However, three critical distinctions emerge in practical settings. First, the true success probabilities are initially uncertain and must be estimated through iterative testing, transforming the problem into a variant of online knapsack optimization. Second, components frequently exhibit non-additive interactions, either positive

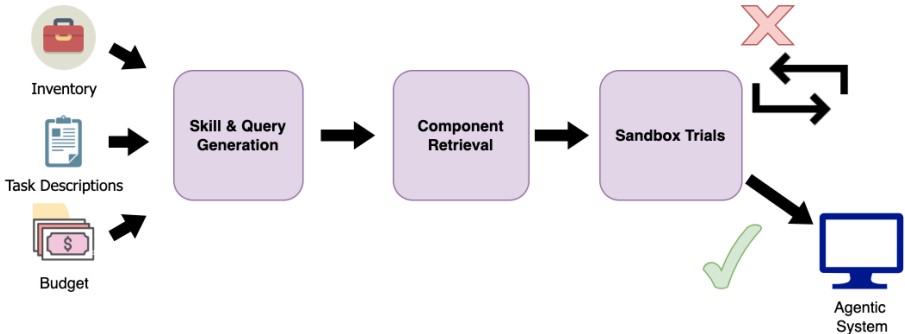

Figure 2: Overview of our proposed online knapsack composer. Similar to offline baselines, the workflow begins with generating skills and queries from the given task descriptions (Appendix A.1). Then, the composer retrieves components from the inventory given skill descriptions. With the retrieved components, the composer tests them individually. If value-to-price ratio meets the online knapsack threshold, then it is added as part of the agentic system. Otherwise, the search continues.

synergies or detrimental conflicts, that may introduce quadratic coupling terms into the objective function. Third, the inventory itself evolves dynamically as new components are added or existing ones are updated, requiring solution methods that support incremental recomputation.

These complications notwithstanding, the knapsack analogy provides both conceptual clarity and algorithmic leverage. Practical applications range from enterprise settings where development teams select from internal tool registries, to marketplace scenarios where automated agents assemble solutions from commercial API inventories. In all cases, the core challenge remains consistent: navigating the combinatorial explosion of possible configurations to identify cost-effective compositions that satisfy stringent performance requirements. The composer agent introduced in Section 4 addresses this challenge through a novel synthesis of constrained optimization and empirical validation techniques.

## 4 Composer Agents: From Semantic Retrieval to Knapsack Selection

A composer can be generalized as a function $f$ that will return a subset $\mathcal{S}$ of components from an inventory $\mathcal{A}$: $f(\mathcal{A}, \tau, B) = \mathcal{S} \subseteq \mathcal{I}$ for given task $\tau$ and budget $B$. Ideally, we would want $f$ to return the optimal solution $S^*$ for equation 1. However, we note that finding this solution is non-trivial due to dynamic behavior of these components and difficulty in assessing their true value. Below, we list and explain our proposed composers.

**Identity Composer:** The first example of a primitive composer is simply to return all components in an inventory where $f(\mathcal{A}, \tau, B) = \mathcal{A}$. We will refer to this as the identity composer, as it resembles an identity function.

**Retrieval Composer:** Next, we should consider taking natural language descriptions into account when designing a more intelligent composer. Recall Section 3 assumes that task $\tau$ has a description and each component $a_i$ in the inventory has a description $d_i$. Many existing works rely on embedding models for semantic retrieval of components like tools (Section 2). However, the challenge here is deciding the query to retrieve from. We also want to make sure that the retrieved components are not redundant. To do so, we augment the retrieval composer with the ability to parse task descriptions into a list of **skills**. Each skill should be a required, core ability for task completion. Table 1 shows examples of generated skills from task descriptions in one of our runs for GAIA with Claude 3.5 Sonnet.

Given task $\tau$ and its description $x$, the retrieval composer will first generate a list of skills $m \in \mathcal{M}$, each with a name and a description. The retrieval composer then queries the inventory with the skill name and description to find the most relevant component. The selected subset $\mathcal{S}$ will be the top-1 retrieval results based on the skills. Note that semantic matching has been used in prior work in agent discovery [5] and more generally in service discovery [20]. Therefore, we include a retrieval-only baseline in our experiments for fair comparison.

Table 1: Examples of skills and test queries that are generated based on GAIA task description with Claude 3.5 Sonnet. The generated information is then used to select each tool for the agent.

| Name | Description | Test Queries |
|---|---|---|
| Web Research | Ability to search the internet, retrieve information, and provide concise web-based research results | 1. Find the current population of Tokyo
2. What is the latest stock price for Apple Inc? |
| Code Assistance | Provide programming support, code generation, debugging, and technical problem-solving across various programming languages | 1. Generate a Python function to calculate Fibonacci sequence
2. Help debug a JavaScript error in a web application |
| Scientific Knowledge | Provide in-depth scientific information, explain complex concepts, and offer research-based insights | 1. Explain the process of photosynthesis
2. Calculate the orbital period of Mars |
| File Management | Handle various file types, perform conversions, extract information, and manage file-related tasks | 1. Convert a PDF document to a Word file
2. Extract text from an image file |
| Quick Information Retrieval | Rapidly provide concise, accurate answers to specific questions across various domains | 1. What is the capital of Australia?
2. How many bones are in the human body? |
| Personal Task Management | Assist with daily personal organization, scheduling, reminders, and lifestyle optimization | 1. Create a meal plan for a week-long diet
2. Schedule a dentist appointment and set a reminder |

**Offline Knapsack Composer:** The above composers do not take budget $B$ into account when making the selection. To find a solution $\mathcal{S}$ with success rate $p_\tau(S)$ close to the optimal success rate $p_\tau(S^*)$ and following the budget constraint, we apply solutions of the Knapsack problem. First, we generate skills based on task description $x$ (Appendix A.1) and retrieve top-$K$ components for each skill. Next, we assign the value of each component based on the similarity score (SIM) used in retrieval. Finally, we use linear programming to find the optimal solution for this multiple-choice knapsack problem [31]:

$$\mathcal{S}_{\text{OFF}} = \arg\max_{\mathcal{S} \subseteq \mathcal{A}} \sum_{a_i \in \mathcal{S}} v_i^{\text{OFF}} \quad \text{s.t.} \quad \begin{cases} \sum_{a_i \in \mathcal{S}} c_i \leq B & \text{(Budget)} \\ \sum_{a_i \in \mathcal{S}} \mathbb{1}[a_i \in \text{TopK}(q)] \geq 1, \forall m \in \mathcal{M} & \text{(Skill coverage)} \end{cases}$$

(2)

where $v_i^{\text{OFF}} = \sum_{m \in \mathcal{M}} \text{SIM}(a_i, m)$.

**Online Knapsack Composer:** Both retrieval and offline knapsack composers only uses semantic retrieval of descriptions to select subset of components. However, descriptions of components may not always be reliable nor aligned with their true capabilities. Components are also prone to changing (e.g. new tool or model updates). Task requirements and environments are also evolving, and there could be "stale" components in the inventory. Thus, we propose an online knapsack composer that iteratively tests the candidate component to assess its true value (Figure 2).

In our problem, the value of the component is not known until it is tested. Therefore, we use the ZCL algorithm [41], a commonly used online knapsack algorithm in the literature. The algorithm assumes there is a lower and upper bound, $L$ and $U$, on the value-to-cost ratio. The solution is theoretically proven to be $\ln(U/L) + 1$-competitive. The algorithm sets a dynamic threshold based on the capacity filled in the knapsack and only accepts an incoming item if its value-to-cost ratio exceeds the threshold $\Psi$. To compute the value, the composer agent has to not only generate the skills but also a set of test questions for each skill (Algorithm 1). The composer agent then uses those questions to evaluate the component. Table 1 shows the test queries that are generated and then used

**Algorithm 1:** Online Knapsack Composer

---

**Input:** Inventory $\mathcal{A}$, task description $x$, budget $B$
**Output:** Selected components $\mathcal{S} \subseteq \mathcal{A}$
$\mathcal{S} \leftarrow \emptyset, \hat{B} \leftarrow B, broken \leftarrow \emptyset$ ;        `// `$\hat{B}$`: remaining budget`
$\mathcal{M} \leftarrow \text{GENERATESKILLS}(x)$ ;     `// {`$(m_j, d_j, Q_j, w_j)$`}: name, desc, queries,`
 `importance`
**foreach** *skill* $m_j \in \mathcal{M}$ **do** $\mathcal{T}_j \leftarrow \text{TOPK}(\mathcal{A}, d_j, K)$;
$L \leftarrow 1 / \max_a c_a, U \leftarrow \sum_j w_j / \min_a c_a$ ;       `// Value-to-cost bounds`
**for** $round \leftarrow 1$ **to** $R$ **do**

> $covered \leftarrow \emptyset$ ;                            `// Reset each round`
> **foreach** *skill* $m_j \in \mathcal{M}$ **do**
>
>> **foreach** *component* $a_i \in \mathcal{T}_j$ **do**
>>
>>> **if** $a_i \in \mathcal{S} \cup broken$ **or** $m_j \in covered$ **or** $c_i > \hat{B}$ **then continue**;
>>> $scores \leftarrow \text{EVALUATE}(a_i, \mathcal{M})$ ;       `// Returns {`$-1, 0, 1$`} per skill`
>>> `// 1: helpful, 0: not helpful, `$-1$`: broken`
>>> **if** $scores[m_j] < 0$ **then**
>>>
>>>> $broken \leftarrow broken \cup \{a_i\}$ ;            `// Never test again`
>>>> **continue**;
>>>
>>> **end**
>>> $v_i \leftarrow \sum_{m_k \notin covered} w_k \cdot \mathbb{1}[scores[m_k] = 1]$ ;     `// Value from new skills`
>>> $\rho_i \leftarrow v_i / c_i$ ;                    `// Value-to-cost ratio`
>>> $z \leftarrow (B - \hat{B})/B, \Psi \leftarrow (\frac{Ue}{L})^z \frac{L}{e}$ ;     `// Fraction spent; ZCL threshold`
>>> **if** $\rho_i \geq \Psi$ **then**
>>>
>>>> $\mathcal{S} \leftarrow \mathcal{S} \cup \{a_i\}, \hat{B} \leftarrow \hat{B} - c_i, covered \leftarrow covered \cup \{m_k : scores[m_k] = 1\}$;
>>>
>>> **end**
>>
>> **end**
>
> **end**

**end**
**return** $\mathcal{S}$

---

to test tools during the sandboxing trials. Appendix A.2 goes over the prompt and details on how the component is judged. The component's value is then based on this assessment.

If a component is successfully equipped, then the composer agent will move forward to the next skill and iterate over the top-K components until it finds a successful one. Note to further optimize upon the runtime of Algorithm 1, we make modifications such as: 1) stopping sandboxing early if component breaks and preventing it from being tested again, 2) once a skill is covered, we do not assess that skill again for other candidate tools. We cover these shortcut optimizations in Appendix A.3 and algorithmic details in Algorithm 2.

## 5 Experiments

This section covers two sets of experiments to evaluate our proposed approaches for agent composition. The first set of experiments look at how to compose single agents through selecting the appropriate tools. The second set of experiments focus on how to compose multi-agent teams through selecting the right, specialized sub-agents. In both sets of experiments, we compose the agents for a particular task. Once the components are selected, we fix the agentic setup and run evaluation on the benchmarks. We explain in detail how the evaluation is conducted for each set of experiments in their respective subsections. In the main body, we present a subset of results and have full results in Appendix A.9.

### 5.1 Composing Single Agents

**Inventory** For single-agent experiments, the inventory $\mathcal{A}$ consists of 120 tools. For this tool inventory, we first collect tools that are actual APIs and can be easily found on Langchain [15], like arXiv, PubMed, SemanticScholar, web search, etc. For the rest of the tools, we collect a small subset from the largest available tool retrieval benchmark, ToolRet [29]. We discover that many tools in

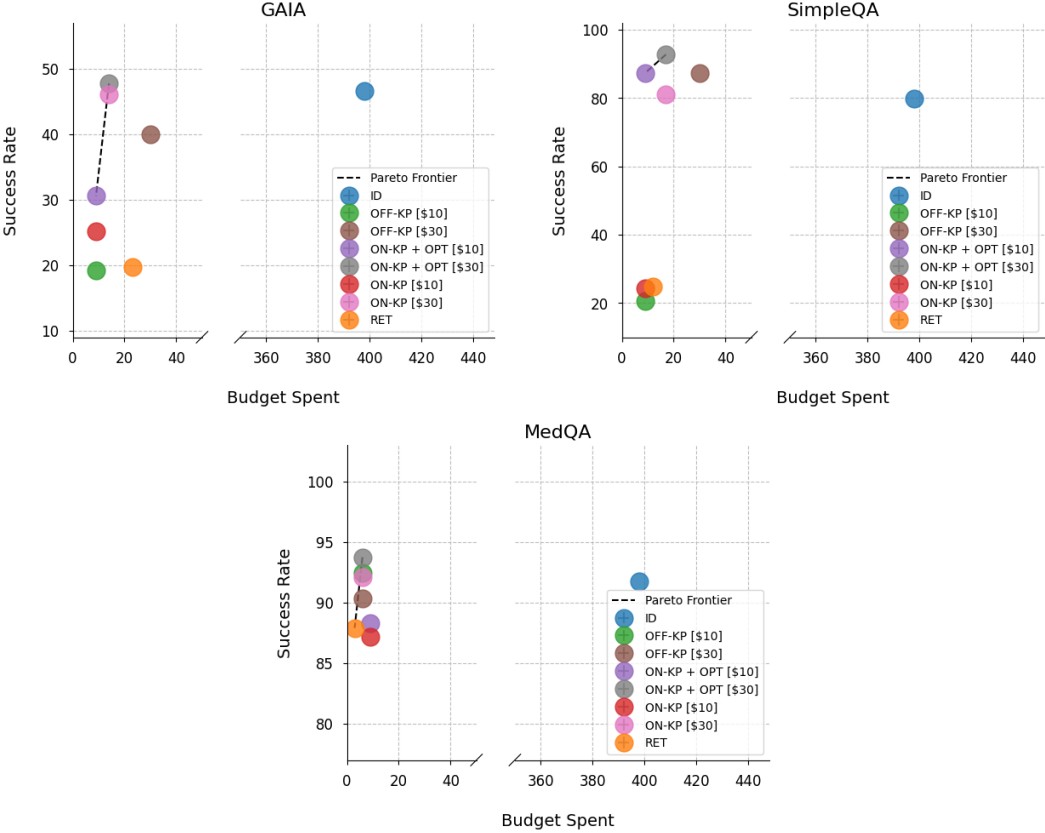

Figure 3: Results of Claude 3.5 Sonnet single-agent experiments where we evaluate all the approaches to select the most relevant tools given task description. After equipping the agent with those tools, we then run evaluation on the dataset and plot success rate against the budget spent on tools. We plot the pareto frontier which represents that no other agent can perform better simultaneously in both success rate and cost. Overall, the **online knapsack with AvaTaR optimization** ($30) shows to be cost-effective and highest performing approach, followed by **online knapsack without optimization** ($30).

this benchmark are not readily available to use and very specialized for one usecase. This is likely because the benchmark is used to test retrieval accuracy. For more details on the construction of the tool inventory, refer to Appendix A.4. For pricing, we estimate the cost of a tool to consist of the input tokens (from the tool schemas) during inference and cost of the API call. The pricing for about 5K calls to the agent for a free API tool is around $3. For the paid APIs, we then estimate the tools to be about $5 and $8. See Appendix A.5 for more details.

**Models**   We fix the agent workflow to CodeAct [33], where the agent is prompted to generate code in order to invoke tools rather than in standardized JSON format [38]. We choose CodeAct as it has shown to outperform other tool-calling frameworks. We experiment with both Claude 3.5 Sonnet (`claude-3-5-sonnet-20241022`), Claude 3.5 Haiku (`claude-3-5-haiku-20241022`) [3], and Claude 3.7 Sonnet (`claude-3-7-sonnet-20250219`). In one experiment, we will use the same model for the composer agent and the candidate agent that is being evaluated. For embedding model, we use BGE-Large-English embeddings (`bge-large-en-v1.5`) [37].

**Datasets**   For single-agent experiments, we evaluate on GAIA, SimpleQA, and MedQA. GAIA [21], which stands for General AI Assistants, is a comprehensive evaluation framework designed to assess the capabilities of AI systems in handling real-world scenarios. It focuses on testing fundamental abilities such as reasoning, multi-modal understanding, web browsing, and tool-use proficiency. Unlike traditional benchmarks that challenge AI with tasks difficult for humans, GAIA poses questions

that are conceptually simple for humans but pose significant challenges for advanced AI models. SimpleQA [34] is a factuality evaluation framework developed by OpenAI to measure the ability of language models to answer short, fact-seeking questions accurately. The questions in SimpleQA are crafted to have a single, indisputable answer, making it easier to evaluate the factual correctness of model responses. MedQA [12] is a comprehensive evaluation framework designed to assess the clinical knowledge of language models in healthcare settings. Questions are derived from the United States Medical License Exams and aims to evaluate the ability of agentic systems to apply medical knowledge in practical scenarios. Note that we reuse the "smolagents" version of GAIA and SimpleQA, which is an active leaderboard for LLM agents hosted on Huggingface [26].

**Approaches** We test on all the composers introduced in Section 4. We pass in the base CodeAct agent, the inventory, task description to the composer. For the knapsack composers, we also pass in the budget, which is set to either $10 or $30 in our experiments. The composer returns a set of tools, which we then equip the CodeAct agent with. We then evaluate the CodeAct agent on the target task. We list the descriptions of each task in Appendix A.7.

For retrieval, we set $K$ to 10. For question generation, we set the number of test questions per skill to 2. During judgement, we ask the composer to judge whether the tool was useful for the agent to answer the question. Additionally, we include an approach where we conduct tool selection through online knapsack and then conduct an additional round of **prompt optimization** using AvaTaR [36]. The feedback for prompt optimization is derived from the tool sandboxing trajectories when the composer has to test the CodeAct agent with the tool. Therefore, we already have existing twelve trajectories per tool for prompt optimization, so we can directly apply AvaTaR. This helps refine the agent's prompt so that it has a better understanding of when to invoke certain tools.

**Results** Figure 3 shows the results from the Claude 3.5 Sonnet experiments where we plot success rate against the budget spent on tools. Following the accuracy-cost analysis done by Kapoor et al. [13], we plot the pareto frontier. Approaches that are on the pareto frontier signify that there is no other method that outperforms them simultaneously in both dimensions. Overall, we observe the highest performing approach to be **online knapsack with AvaTaR optimization** ($30 budget constraint). On all three datasets, its success rate is higher than **identity** but cost is much lower. Similarly, **online knapsack without AvaTaR optimization** shows to be on the pareto frontier for GAIA and MedQA, and has relatively higher success rate than the other approaches. Retrieval-only approaches (**retrieval** and **offline knapsack**) tend to fare worse across all three datasets, which validates previous findings that only using retrieval for tool discovery is insufficient [29].

## 5.2 Composing Multi-Agent Systems

**Setup** For multi-agent benchmarking, we use the end-to-end multi-agent evaluation framework from Shu et al. [30]. Their framework assumes a hierarchical multi-agent collaboration (MAC) architecture where one agent acts as a supervisor and delegates tasks to specialist sub-agents. The benchmarking framework publicly released a MAC benchmarking dataset that includes an inventory of about 20 sub-agents for a few enterprise domains. The evaluation involves using the scenario from the dataset as input and then simulating the conversation between the user and multi-agent team. If the team's trajectory satisfies the annotated list of assertions, then it is considered a success.

We set up ReAct tool-calling agents as done in the paper [38] and benchmark on two domains: travel and mortgage. We further synthetically augment the original inventory of agents to about 117 agents (Appendix A.6). We arbitrarily set the price of each sub-agent to $1. All sub-agents use the same underlying model and their tools are being simulated in this setup. In this controlled setting, there is no meaningful variation in runtime or API costs between sub-agents, so assigning a uniform cost allows us to isolate and evaluate the impact of agent selection strategies, making it feasible to compare how this setting differs from Section 5.1 where the pricing was more varied across items. For these experiments, we fix the budget to $3 and $6.

For our experiments, we first pass in task descriptions (Appendix A.7), agent inventory, and optional budget to the composer to select the most relevant sub-agents. Then, we set up multi-agent team where the supervisor has the original profile from [30] but with newly selected sub-agents from agent composition. Finally, we simulate the interactions between user and the newly formed MAC team.

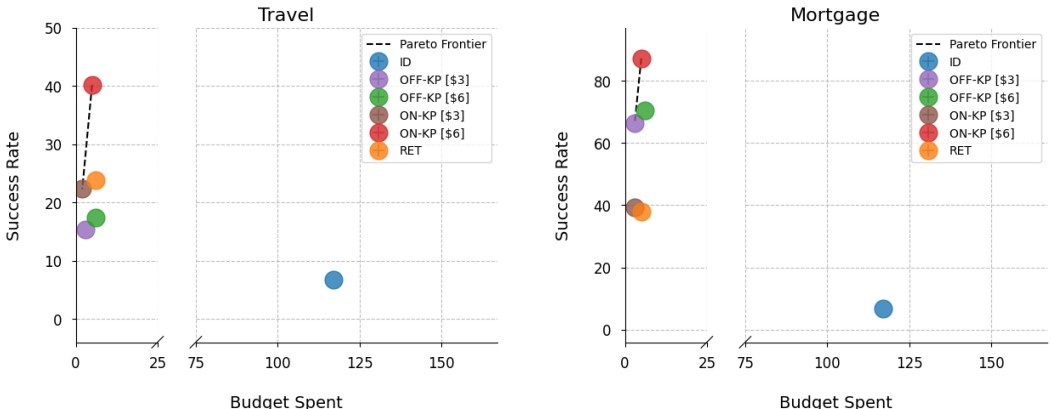

Figure 4: Results of Claude 3.5 Sonnet multi-agent experiments where we evaluate all approaches to select most relevant tools given task description. After equipping the agent with those tools, we then run evaluation on the dataset and plot success rate against tool costs. Overall, **online knapsack** shows to be cost-effective and highest performing approach.

**Results**  Figure 4 shows the results from the multi-agent experiments. We plot overall goal success rate against budget spent on the agent selection. In both domains, **online knapsack** ($6) is the highest-performing approach on the pareto frontier. Unlike single-agent experiments, **identity** no longer shows high success rate as supervisor agent has trouble delegating task to an inventory of more than 100 diverse agents. Possibly, an improved multi-agent framework could be an alternative approach to handle large-scale network of agents [5].

## 6 Discussion

**Improvements with online knapsack in single agent composition.** In Table 2, we show the tools that are chosen during the experiments to compare the various composers. For GAIA and SimpleQA, having "web search" tool is critical as the questions revolve around more recent or esoteric topics. For **retrieval composer**, we do not see any "web search" tools being picked for GAIA and SimpleQA. The tool selection highly depends on the ability of the retriever to match skill descriptions to tool descriptions. While "Web Browsing" was a skill generated for GAIA, the description for this skill seemed to match closely with the "get_article_content" where both mentioned web pages. However, "get_article_content" is not a tool to help search the web. **Offline knapsack** was able to choose "web_search_free" which is the web search tool that is cheaper but with severe throttling limits. **Online knapsack** chose "web_search_paid", which is the most appropriate web search tool for these tasks. Since the **online knapsack** composer can also test out these tools, it can detect that "web_search_paid" can effectively search the web. This explains the higher success rate that we see with **online knapsack** in Figure 3.

We additionally observe that the prompt optimization helps to boost performance for SimpleQA. We attribute this improvement to the revised agent system prompt offering clear guidelines on query formulation for effective search, tool usage guidelines, source verification, and enhanced error-recovery protocol. For details on the impact of adaptation for tool use, please refer to Appendix A.8.

Appendix A.9 shows results on several models, including ones from Llama 4 [1] and Qwen 2.5 [32] model families. We observe similar results where **online knapsack** tends to outperform the baselines. Furthermore, we observe consistency of these results across multiple runs (Table 12) .

**Improvements with online knapsack in multi-agent composition.** For multi-agent experiments, we observe the retrieval-based methods would often select the "distractor" agents that we add to the inventory. These are agents that overlap in semantic description with the original MAC agents that should have chosen but lack the tools and proper instructions to carry out the tasks successfully. For instance, **offline knapsack** composer ($6) chose five agents that lacked true capabilities for travel, which was why it had performed worse for this domain (Figure 4). On the other hand,

Table 2: Tools that are selected by the various composers over GAIA and SimpleQA. Retrieval composer tends to miss out on relevant tools for the task. Offline knapsack often includes too many irrelevant tools. Online knapsack seems to have both precise and well-covered choice of tools.

| Composer | GAIA | SimpleQA |
|---|---|---|
| Retrieval | pub_med, read_file, wolfram_alpha, job_title_autocomplete, get_article_content, number_fact | wikipedia, pub_med, sources, number_fact |
| Offline Knapsack ($30) | web_search_free, arxiv, wikipedia, pub_med, read_file, semanticscholar, instantaneous_values_service, get_recordings, get_article_content, number_fact | web_search_free, wikipedia, pub_med, semanticscholar, query_by_id, standard_language_detection, get_recordings, get_article_content, symbols_faq, number_fact |
| Online Knapsack ($30) | web_search_paid, arxiv, wikipedia | web_search_paid, web_search_free, wikipedia, semanticscholar |

**online knapsack** consistently would avoid these "distractor" agents as it was obvious that they were non-operable.

**Limitations.** First, our problem definition (Section 3) clearly states that we must assume the task is well-defined with a clear description. This applies to many scenarios where developers already have a clear understanding of their architectural goals and needs. However, there still exists real-life usecases where developers have ambiguous goals and require more exploration. Future work could look at further developing our approach for tasks with unclear specifications. Second, there could be alternative agent composition approaches to compare against, ranging from simple (greedy approach) to more complex approaches such as formulating agent composition as a Markov Decision Process. Additionally, we could also try testing combinations of agentic components rather than individually. While **online knapsack** composer outperform all baselines, the sandbox trials does take additional time (10-30 minutes depending on budget). A more optimized approach can look towards reducing it. Third, our prompt optimization results show regression in a few settings. This emphasizes the need for a more robust optimization method. Finally, agent composition has many positive impacts contributing to rapid creation of AI systems, but there could be potential negative effects. Without careful monitoring of the creation of agentic systems, malicious tools or other components could possibly be added to the system.

## 7   Conclusion

We formalize the problem of selecting existing agentic components as agent composition. We reduce agent composition to a knapsack problem to optimize selection of components for success rate within a constrained budget. We set up two sets of experiments, one on selecting tools for a single agent and the other for composing multi-agent systems with existing sub-agents. Online knapsack composer can optimize composition because it not only uses semantic search to discover components but also tests them to assess their real-time utility. Through application of online knapsack algorithm, we can efficiently determine whether to select a component based on its assessed value-to-price ratio. Future works on agent composition may look towards more dynamic methods, such as learning how to compose from past experience.

## Acknowledgments and Disclosure of Funding

This work has been supported and funded by Amazon Web Services throughout the course of the project. Michelle Yuan is currently affiliated with Oracle but completed this work during her time at Amazon Web Services. The views and conclusions expressed in this paper are those of the authors and do not reflect the views, policies, or positions of Amazon, Oracle, or their affiliates. We thank Nilaksh Das, Etsuko Ishii, Raphael Shu, Daniele Bonadiman, Tamer Alkhouli, Katerina Margatina, and Salvatore Romeo for their valuable insights.

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

## A Appendix

### A.1 Skill and Query Generation Prompt

For both offline and online knapsack composers, skill generation is a critical component of the workflow. Note that we combine skill and query generation in one step to reduce the steps in the composition workflow. Below is the prompt used for the composer agent to generate skills and queries. We also ask the composer to generate reference plans for each query, which will help with the judgement part (Appendix A.2).

```
You are a task parser.  Your job is to parse the following tasks into
a fine-grained list of no more than six skills.  Each skill should
be distinct and unique.  For each skill, you also must provide a
description, an importance value rated on a scale of 1 to 10, and a
list of 3 test queries with plans.  Queries should provide all the
context needed for an agent to solve the task.  Queries should be
designed so that agent must use a tool to solve, but should NOT take
more than 1 step or tool call.  Plans should describe how an agent
should solve the query, which are a short list of actions that the
agent needs to take or high-level reasoning steps that the agent needs
to think about.
```

The output schema for a skill is as follows:

```
{
    name: str
    importance: int
    description: str
    queries: list[{query: str, plan: str}]
}
```

### A.2 Prompts for Sandbox Judgements

During the sandbox trials (Figure 2), the composer has to judge whether the agentic component is useful for completing the task. We set up two variations of the prompts, one for single agent and another for multi-agent experiments. In the single agent experiments, the composer needs to determine whether the candidate tool has helped the agent achieve the goal. In the multi-agent experiments, the composer needs to determine whether the candidate sub-agent has helped accomplish the user goals.

Below is the prompt for the sandbox judgement in single-agent experiments:

```
You will be given a query, reference plan, the tool's name, and agent's
steps to answering the query.  You are to judge whether the tool has
helped the agent in answering the query and whether the tool is broken.
If the tool appears to have errors after multiple attempts or have
potentially harmful effects on the environment, you must return True
for "broken".  If the agent was able to answer the query without using
the tool, you must return False for "helpful".  Only return True for
"helpful" if the tool helped the agent answer the query.

Query: {query}
Reference Plan: {reference plan}
Tool: {tool}
Agent's Steps: {agent steps}
```

Below is the prompt for the sandbox judgement in multi-agent experiments:

```
You will be given a query, reference plan, and agent's steps to
answering the query.  You are to judge whether the agent is helpful
in answering the query and whether the agent is broken.  If the agent
is able to call the right tools and answer the query using information
from the tool calls, you must return True for "helpful".  If the
agent does not complete the task but explains that they cannot do
so, just return "False" for "helpful" and "False" for "broken".  ONLY
return True for broken if the agent is toxic, has potentially harmful
effects on the environment, completely irrelevant for the task.  Also
return True for broken if the agent tries to call tools that are not
available.

Query: {query}
Reference Plan: {reference plan}
Tool: {tool}
Agent's Steps: {agent steps}
```

Note that the output schema for both experiments is the same:

```
{
    helpful: bool
    broken: bool
    reason: str
}
```

### A.3 Modifications to ZCL Algorithm for Online Knapsack Composer

In Algorithm 1, we show how we use ZCL algorithm for the online knapsack composer. We make some minor modifications to reduce the number of sandbox trials:

- **Early stopping for sandboxing:** In preliminary development, we observe slowdowns caused by repeated testing of broken components, which could have been avoided by just immediately flagging these components as broken. We added an additional judgement in the prompt (Appendix A.2) to flag these as broken. Once a component is flagged as broken, we end the sandboxing and prevent it from being tested again.

- **Skipping covered skills:** During sandboxing, we keep track of the skills that are covered. In other words, when a component passes the judge's assessment for a skill and the component is equipped, then the skill is recorded as "covered". Once that skill is covered, then we no longer need to find more components relevant to that skill. This not only reduces the sandboxing trials but also prevents equipping components that overlap in skill.

We disclose an extended version of the algorithm in Algorithm 2.

**Algorithm 2:** Online Knapsack Composer - Detailed

---

**Input:** Inventory $\mathcal{A}$, task description $x$, budget $B$
**Output:** Selected components $\mathcal{S} \subseteq \mathcal{A}$
```
// Initialization
```
$\mathcal{S} \leftarrow \emptyset, \hat{B} \leftarrow B, broken \leftarrow \emptyset$ ;          `// B̂:  remaining budget`
```
// Generate skills from task description
```
$\mathcal{M} \leftarrow \text{GENERATESKILLS}(x)$ ;      `// {(m_j, d_j, Q_j, w_j)}:  name, desc, queries,`
`importance`
```
// Retrieve top-K candidates per skill
```
**foreach** *skill* $m_j \in \mathcal{M}$ **do**
  |   $\mathcal{T}_j \leftarrow \text{TOPK}(\mathcal{A}, d_j, K)$;
**end**
```
// Compute value-to-cost ratio bounds
```
$L \leftarrow 1 / \max_{a \in \mathcal{A}} c_a$ ;            `// Worst case:  max cost, min value`
$U \leftarrow \sum_j w_j / \min_{a \in \mathcal{A}} c_a$ ;          `// Best case:  min cost, max value`
```
// Iterative online knapsack rounds
```
**for** $round \leftarrow 1$ **to** $R$ **do**
  |   $covered \leftarrow \emptyset$ ;                 `// Reset covered skills each round`
  |   **foreach** *skill* $m_j \in \mathcal{M}$ **do**
  |    |   **foreach** *component* $a_i \in \mathcal{T}_j$ **do**
  |    |    |   **if** $a_i \in \mathcal{S} \cup broken$ **or** $m_j \in covered$ **or** $c_i > \hat{B}$ **then**
  |    |    |    |   **continue** ;             `// Skip if processed or infeasible`
  |    |    |   **end**
  |    |    |   $scores \leftarrow \text{TESTONQUERIES}(a_i, \mathcal{M})$ ;     `// Test on all skill queries`
  |    |    |   **if** $scores[m_j] < 0$ **then**
  |    |    |    |   $broken \leftarrow broken \cup \{a_i\}$ ;           `// Mark as broken`
  |    |    |    |   **continue**;
  |    |    |   **end**
  |    |    |   $v_i \leftarrow \sum_{m_k \notin covered} w_k \cdot \mathbb{1}[scores[m_k] = 1]$ ;    `// Value from new skills`
  |    |    |   $\rho_i \leftarrow v_i / c_i$ ;                 `// Value-to-cost ratio`
  |    |    |   $z \leftarrow (B - \hat{B}) / B$ ;          `// Fraction of budget spent`
  |    |    |   $\Psi \leftarrow \left(\frac{Ue}{L}\right)^z \frac{L}{e}$ ;            `// ZCL threshold`
  |    |    |   **if** $\rho_i \geq \Psi$ **then**
  |    |    |    |   $\mathcal{S} \leftarrow \mathcal{S} \cup \{a_i\}$;
  |    |    |    |   $\hat{B} \leftarrow \hat{B} - c_i$;
  |    |    |    |   $covered \leftarrow covered \cup \{m_k : scores[m_k] = 1\}$;
  |    |    |   **end**
  |    |   **end**
  |   **end**
**end**
**return** $\mathcal{S}$

---

## A.4   Tool Inventory

We created the tool inventory from Langchain tools and the ToolRet dataset [28]. The tools can be divided into 23 distinct categories based on their functional similarities, such as "personal task assistance", "fact retrieval", and "financial information". The category with the highest number of tools is "personal task assistance", which includes 24 tools. Conversely, categories like "arithmetic calculations," "health information," and "event management" each contain only one tool, indicating a more specialized focus in these areas.

A detailed overlap analysis revealed that out of the 120 unique tools, 35 tools (28.5%) appear in multiple categories, demonstrating a high degree of functional versatility. The degree of overlap was further categorized into high (4+ categories), medium (3 categories), and low (2 categories). Notably, tools like wikipedia, pubmed, and semanticscholar, which are primarily information retrieval

tools, appear in multiple categories related to knowledge retrieval, fact-checking, and reference. This highlights the dominance of information retrieval tools in the inventory.

## A.5 Pricing of Components

For single-agent experiments, we have an inventory of tools that is a mix of free, free with rate limits, and paid APIs. Our pricing model was primarily based on two costs: 1) cost of API, 2) cost of its schema in terms of input tokens. To simplify the pricing model, we assume cost of paid API to be $5 per 5000 queries (based on SerperAPI search pricing) and average tool schema token count of 200. If Claude API is priced at $3 per million input tokens, then this translates to a cost of $3 for 5000 queries with the tool schema. Therefore, the estimated cost of a paid tool would be $8 per 5000 queries. For a free tool, it would be $3 per 5000 queries, as it only needs to consider the cost of the tool schema in the input tokens. We then assign $5 to tools that are free but up to a certain number of queries.

## A.6 Agent Inventory

We synthetically generate new agents to span diverse domains unrelated to the original MAC domains of mortgage, travel, and software, including healthcare (e.g., appointment scheduler agent, telemedicine connector agent), e-commerce (e.g., product search agent, price tracker agent), government/legal (e.g., tax filing agent, legal consultation agent), education (e.g., homework help agent, course recommender agent), career/jobs (e.g., resume builder agent, salary insights agent), media/entertainment (e.g., movie recommender agent, news summarizer agent), social media/messaging (e.g., toxic comment filter agent, follower growth agent), and sustainability/lifestyle (e.g., carbon footprint agent, recycling guide agent). We also include ten agents that overlapped with the original MAC agents in profile descriptions but had no tools attached to them. This would assess whether composition could discover the agents that were truly capable of accomplishing the user goals in MAC.

## A.7 Task Descriptions

Table 3 has the task descriptions for each dataset. These task descriptions are part of the input to the agent composition framework.

## A.8 Impact of Prompt Optimization on Tool Use

This appendix explains the impact of prompt optimization. We illustrate the improvement with a Claude 3.5 Sonnet benchmarking run for SimpleQA. The goal of optimizing the system prompt is to improve the agent's tool selection, reasoning transparency, and error recovery, which led to notable gains in both answer accuracy and operational efficiency.

Initially, the system prompt provided only general guidance and lacked explicit structure for decomposing tasks, matching tools to subtasks, and recovering from tool failures. As a result, agents sometimes performed redundant or suboptimal tool calls, unnecessarily repeated queries, and often failed to provide answers when information was not immediately available.

Prompt optimization introduces more systematic instructions: the agent is now required to break each question down into sub-tasks, select the most appropriate tool for each sub-task, and avoid redundant tool calls. Specific preferences are introduced for tool choice. For example, using `wikipedia` for general knowledge and `semanticscholar` for scientific queries. The revised prompt also implements better query reformulation, result validation, and error recovery, producing a more disciplined, efficient, and interpretable agent behavior.

Note that the prompt optimization is done during the agent composition phase where the input feedback is based on the recorded sandboxing trials. The feedback signals for prompt optimization is given by the same LLM-as-a-judge that helps score whether the tool is useful for the agent on the given task. After prompt optimization, the agent's system prompt is fixed for benchmarking, which is the same as the other approaches.

Table 3: Table of Datasets and Task Descriptions

| Dataset | Task Description |
|---------|-----------------|
| SimpleQA | - Answer short, factual questions that should have a single correct answer
- Answers to the questions may require searching the web
- Questions should be from a wide range of topics,
including TV shows, music, sports, geography, art, science, technology, etc. |
| GAIA | - Help user with tasks that need web browsing, coding, or filetype handling
- Help user with various assistant use cases,
such as daily personal tasks, science, or general knowledge
- Answer questions that should have a short, single correct answer |
| MedQA | - Helps users with medical questions
that need web browsing for medical or clinical information,
reviewing published medical articles and other medical texts
- Answer medical questions from US medical licensing exams (USMLE)
- Topics cover various medical domains
including anatomy, physiology, pharmacology, and clinical practice |
| Travel (MAC) | - Help users book flights, accommodations,
and car rentals for their upcoming trips
- Help users with any queries regarding travel planning
like local attractions or weather forecast |
| Mortgage (MAC) | - Help users with mortgage planning and financing
- Help users find property information
- Help users retrieve their banking and credit information |

**Example Query: "What year did Reinhold Rudenberg retire?"**

The following logs illustrate the agent's behavior before and after prompt optimization in response to a query from the SimpleQA dataset.

**Before Prompt Optimization**

```
Thought: I'll search for information about Reinhold Rudenberg.
Tool Call 1: wikipedia(''Reinhold Rudenberg'')
→ Observation: No retirement year mentioned.

Thought: Try web search.
Tool Call 2:
web_search_free(''Reinhold Rudenberg career Harvard retirement'')
→ Error: Rate-limited.

Tool Call 3: semanticscholar(''Reinhold Rudenberg Harvard'')
→ Observation: Retrieved unrelated academic papers.

Thought: Try again with web search.
Tool Call 4: web_search_free(...)
→ Error: Rate-limited again.

Final Decision: ''Cannot determine retirement year.''
```

**After Prompt Optimization**

```
Thought: Begin with general biography search.
Tool Call 1: wikipedia(''Reinhold Rudenberg'')
→ Observation: Biographical overview, no retirement data.
```

```
Thought: Try academic source for specific timeline.
Tool Call 2: semanticscholar(''Reinhold Rudenberg retirement'')
→ Observation: Found timeline-aligned publication data.

Thought: Confirm with direct query.
Tool Call 3: web_search_paid(''When did Reinhold Rudenberg retire'')
→ Observation: Found retirement year: 1952

Tool Call 4: final_answer(1952)
```

Prompt optimization produces measurable improvements: after refinement, the agent answers correctly, reduces unnecessary tool usage, avoids repetitive errors, and presents a clearer chain of reasoning. Task breakdown, explicit tool assignment, and robust fallback strategies together improves not only answer accuracy but also the efficiency and interpretability of the agent's response.

## A.9 Detailed Performance Results

In this section, we report the quantitative results for the Claude 3.5 Sonnet, Claude 3.5 Haiku, Claude 3.7, Sonnet, Qwen 2.5 72B, Llama 3.3 70B, Llama 4 Maverick, and Llama 4 Scout. The plots in Fig. 3 correspond to the results in Table 8. In the last table, we have multiple runs of the SimpleQA experiments and show minimal variance across three runs.

Table 4: Performance across approaches, domains, and budget constraints using Claude 3.5 Sonnet. We used different colors to denote the highest-performing experiment for GAIA (red), SimpleQA (blue), and MedQA (violet).

| Total budget constraint | Domain | Approach | Success Rate (avg) | No. tools used | Budget spent |
|---|---|---|---|---|---|
| **Baselines** **Budget: NA** | | | | | |
| | GAIA | Identity | **0.47** | 122 | 398 |
| | | Top-1 retrieval | 0.19 | 6 | 23 |
| | SimpleQA | Identity | 0.80 | 122 | 398 |
| | | Top-1 retrieval | 0.24 | 4 | 12 |
| | MedQA | Identity | 0.92 | 122 | 398 |
| | | Top-1 retrieval | 0.87 | 1 | 3 |
| **Knapsack Composers** **Budget: $10** | | | | | |
| | GAIA | Offline knapsack | 0.19 | 3 | 9 |
| | | Online knapsack | 0.25 | 3 | 9 |
| | | Online knapsack + OPT | 0.31 | 3 | 9 |
| | SimpleQA | Offline knapsack | 0.24 | 3 | 9 |
| | | Online knapsack | 0.24 | 3 | 9 |
| | | Online knapsack + OPT | 0.82 | 3 | 9 |
| | MedQA | Offline knapsack | 0.87 | 2 | 6 |
| | | Online knapsack | 0.91 | 2 | 6 |
| | | Online knapsack + OPT | 0.89 | 3 | 9 |
| **Knapsack Composers** **Budget: $30** | | | | | |
| | GAIA | Offline knapsack | 0.41 | 10 | 30 |
| | | Online knapsack | **0.47** | 4 | 12 |
| | | Online knapsack + OPT | 0.47 | 4 | 14 |
| | SimpleQA | Offline knapsack | 0.88 | 10 | 30 |
| | | Online knapsack | 0.92 | 4 | 12 |
| | | Online knapsack + OPT | **0.92** | 4 | 12 |
| | MedQA | Offline knapsack | 0.91 | 3 | 9 |
| | | Online knapsack | 0.93 | 2 | 6 |
| | | Online knapsack + OPT | **0.93** | 2 | 6 |

Table 5: Performance comparison across approaches, domains, and budget levels for multi-agent composition using Claude 3.5 Sonnet for both primary and secondary agents. We highlight the best result for travel domain in blue, and in red for the mortgage domain.

| Total budget constraint | Domain | Approach | Overall GSR | Partial GSR | Budget spent | Run duration (avg sec) |
|---|---|---|---|---|---|---|
| **Baselines** **Budget:NA** | | | | | | |
| | Travel | Identity | 0.07 | 0.09 | 17 | 30.04 |
| | | Top-1 retrieval | 0.23 | 0.57 | 6 | 30.24 |
| | Mortgage | Identity | 0.07 | 0.02 | 117 | 7.27 |
| | | Top-1 retrieval | 0.37 | 0.68 | 5 | 13.22 |
| **Knapsack Composers** **Budget: $3** | | | | | | |
| | Travel | Offline knapsack | 0.16 | 0.34 | 3 | 17.77 |
| | | Online knapsack | 0.23 | 0.53 | 3 | 26.42 |
| | Mortgage | Offline knapsack | 0.67 | 0.81 | 3 | 24.91 |
| | | Online knapsack | 0.40 | 0.73 | 3 | 17.28 |
| **Knapsack Composers** **Budget: $6** | | | | | | |
| | Travel | Offline knapsack | 0.17 | 0.38 | 3 | 16.08 |
| | | Online knapsack | **0.40** | 0.69 | 5 | 33.93 |
| | Mortgage | Offline knapsack | 0.70 | 0.89 | 3 | 20.30 |
| | | Online knapsack | **0.87** | 0.93 | 5 | 24.13 |

Table 6: Performance across approaches, domains, and budget constraints using Claude 3.7 Sonnet. We used different colors to denote the highest-performing experiment for GAIA (red), SimpleQA (blue), and MedQA (violet). We used the same colors but highlighted in bold to denote the highest performing experiment setting for each of the three domains.

| Total budget constraint | Domain | Approach | Success Rate (avg) | No. tools used | Budget spent | Run duration (avg sec) |
|---|---|---|---|---|---|---|
| **Baselines** **Budget: NA** | | | | | | |
| | GAIA | Identity | 0.47 | 122 | 398 | 100.1 |
| | | Top-1 retrieval | 0.25 | 2 | 6 | 106.0 |
| | SimpleQA | Identity | 0.76 | 122 | 398 | 30.4 |
| | | Top-1 retrieval | 0.26 | 2 | 6 | 45.0 |
| | MedQA | Identity | 0.92 | 122 | 398 | 48.0 |
| | | Top-1 retrieval | 0.91 | 1 | 3 | 46.3 |
| **Knapsack Composers** **Budget: $10** | | | | | | |
| | GAIA | Offline knapsack | 0.31 | 3 | 9 | 105.1 |
| | | Online knapsack | **0.50** | 3 | 9 | 75.2 |
| | SimpleQA | Offline knapsack | 0.26 | 3 | 9 | 52.4 |
| | | Online knapsack | **0.92** | 3 | 9 | 29.0 |
| | MedQA | Offline knapsack | 0.91 | 3 | 9 | 64.2 |
| | | Online knapsack | 0.91 | 1 | 3 | 34.1 |
| **Knapsack Composers** **Budget: $30** | | | | | | |
| | GAIA | Offline knapsack | **0.50** | 10 | 30 | 85.8 |
| | | Online knapsack | 0.44 | 10 | 11 | 89.7 |
| | SimpleQA | Offline knapsack | 0.80 | 10 | 30 | 22.9 |
| | | Online knapsack | **0.92** | 4 | 12 | 22.9 |
| | MedQA | Offline knapsack | 0.89 | 3 | 9 | 49.7 |
| | | Online knapsack | **0.93** | 1 | 3 | 30.3 |

Table 7: Performance of Haiku 3.5 across domains, approaches, and budget constraints for single-agent composition. We used different colors to denote the highest-performing experiment for GAIA (red), SimpleQA (blue), and MedQA (violet).

| Total budget constraint | Domain | Approach | Success Rate (avg) | No. tools used | Budget spent (total tool price) | Run duration (avg sec) |
|---|---|---|---|---|---|---|
| **Baselines** Budget: NA | | | | | | |
| | GAIA | Identity | 0.34 | 122 | 398 | 31.8 |
| | | Top-1 retrieval | 0.19 | 5 | 20 | 30.9 |
| | SimpleQA | Identity | 0.80 | 122 | 398 | 15.0 |
| | | Top-1 retrieval | 0.16 | 1 | 3 | 42.5 |
| | MedQA | Identity | 0.83 | 122 | 398 | 15.4 |
| | | Top-1 retrieval | 0.81 | 1 | 3 | 16.3 |
| **Knapsack Composers** Budget: $10 | | | | | | |
| | GAIA | Offline knapsack | 0.16 | 3 | 9 | 46.8 |
| | | Online knapsack | 0.41 | 1 | 8 | 21.5 |
| | | Online knapsack + OPT | 0.41 | 1 | 8 | 24.8 |
| | SimpleQA | Offline knapsack | 0.58 | 3 | 9 | 24.7 |
| | | Online knapsack | 0.16 | 3 | 9 | 15.1 |
| | | Online knapsack + OPT | 0.18 | 3 | 9 | 55.5 |
| | MedQA | Offline knapsack | 0.80 | 3 | 9 | 17.7 |
| | | Online knapsack | 0.82 | 3 | 9 | 15.7 |
| | | Online knapsack + OPT | 0.84 | 3 | 9 | 14.9 |
| **Knapsack Composers** Budget:$30 | | | | | | |
| | GAIA | Offline knapsack | 0.13 | 8 | 29 | 1403.8 |
| | | Online knapsack | 0.47 | 5 | 20 | 23.8 |
| | | Online knapsack + OPT | 0.44 | 5 | 20 | 30.1 |
| | SimpleQA | Offline knapsack | 0.74 | 9 | 29 | 22.7 |
| | | Online knapsack | 0.78 | 4 | 22 | 14.3 |
| | | Online knapsack + OPT | 0.86 | 4 | 22 | 13.7 |
| | MedQA | Offline knapsack | 0.79 | 6 | 20 | 20.3 |
| | | Online knapsack | 0.79 | 5 | 20 | 12.3 |
| | | Online knapsack + OPT | 0.80 | 5 | 20 | 14.6 |

Table 8: Performance across approaches, domains, and budget constraints using Qwen 2.5 72B Instruct. We used different colors to denote the highest-performing experiment for GAIA (red), SimpleQA (blue), and MedQA (violet).

| Total budget constraint | Domain | Approach | Success Rate (avg) | No. tools used | Budget spent |
|---|---|---|---|---|---|
| **Baselines** Budget: NA | | | | | |
| | GAIA | Identity | 0.28 | 122 | 398 |
| | SimpleQA | Identity | 0.86 | 122 | 398 |
| | MedQA | Identity | 0.9 | 122 | 398 |
| **Knapsack Composers** Budget: $10 | | | | | |
| | GAIA | Offline knapsack | 0.20 | 3 | 9 |
| | | Online knapsack | 0.25 | 3 | 9 |
| | SimpleQA | Offline knapsack | 0.58 | 3 | 9 |
| | | Online knapsack | 0.78 | 3 | 9 |
| | MedQA | Offline knapsack | 0.82 | 2 | 6 |
| | | Online knapsack | 0.91 | 2 | 6 |
| **Knapsack Composers** Budget: $30 | | | | | |
| | GAIA | Offline knapsack | 0.22 | 10 | 30 |
| | | Online knapsack | 0.30 | 4 | 12 |
| | SimpleQA | Offline knapsack | 0.79 | 10 | 30 |
| | | Online knapsack | 0.89 | 3 | 14 |
| | MedQA | Offline knapsack | 0.89 | 3 | 9 |
| | | Online knapsack | 0.92 | 1 | 3 |

Table 9: Performance across approaches, domains, and budget constraints using Llama 3.3 70B Instruct. We used different colors to denote the highest-performing experiment for GAIA (red), SimpleQA (blue), and MedQA (violet).

| Total budget constraint | Domain | Approach | Success Rate (avg) | No. tools used | Budget spent |
|---|---|---|---|---|---|
| **Baselines**
**Budget: NA** | | | | | |
| | GAIA | Identity | 0.31 | 122 | 398 |
| | SimpleQA | Identity | 0.78 | 122 | 398 |
| | MedQA | Identity | 0.84 | 122 | 398 |
| **Knapsack Composers**
**Budget: $10** | | | | | |
| | GAIA | Offline knapsack | 0.24 | 3 | 9 |
| | | Online knapsack | 0.27 | 3 | 9 |
| | SimpleQA | Offline knapsack | 0.63 | 3 | 9 |
| | | Online knapsack | 0.71 | 3 | 9 |
| | MedQA | Offline knapsack | 0.74 | 2 | 6 |
| | | Online knapsack | 0.77 | 2 | 6 |
| **Knapsack Composers**
**Budget: $30** | | | | | |
| | GAIA | Offline knapsack | 0.29 | 8 | 30 |
| | | Online knapsack | **0.35** | 4 | 14 |
| | SimpleQA | Offline knapsack | 0.75 | 8 | 30 |
| | | Online knapsack | **0.82** | 4 | 14 |
| | MedQA | Offline knapsack | 0.81 | 4 | 14 |
| | | Online knapsack | **0.88** | 3 | 11 |

Table 10: Performance across approaches, domains, and budget constraints using Llama 4 Maverick 17B Instruct. We used different colors to denote the identity (all tool) results for GAIA (red), SimpleQA (blue), and MedQA (violet). Bolded values indicate the best performing experiment per domain.

| Total budget constraint | Domain | Approach | Success Rate (avg) | No. tools used | Budget spent |
|---|---|---|---|---|---|
| **Baselines**
**Budget: NA** | | | | | |
| | GAIA | Identity (all tools) | 0.38 | 122 | 398 |
| | SimpleQA | Identity (all tools) | **0.82** | 122 | 398 |
| | MedQA | Identity (all tools) | 0.85 | 122 | 398 |
| **Knapsack Composers**
**Budget: $10** | | | | | |
| | GAIA | Offline knapsack | 0.19 | 3 | 9 |
| | | Online knapsack | 0.19 | 3 | 9 |
| | SimpleQA | Offline knapsack | 0.28 | 3 | 9 |
| | | Online knapsack | 0.78 | 3 | 9 |
| | MedQA | Offline knapsack | 0.88 | 3 | 9 |
| | | Online knapsack | **0.91** | 3 | 9 |
| **Knapsack Composers**
**Budget: $30** | | | | | |
| | GAIA | Offline knapsack | 0.31 | 10 | 30 |
| | | Online knapsack | **0.44** | 5 | 20 |
| | SimpleQA | Offline knapsack | 0.78 | 10 | 30 |
| | | Online knapsack | 0.78 | 2 | 11 |
| | MedQA | Offline knapsack | 0.89 | 7 | 21 |
| | | Online knapsack | 0.89 | 4 | 12 |

Table 11: Performance across approaches, domains, and budget constraints using Llama 4 Scout 17B Instruct. We used different colors to denote the highest-performing experiment for GAIA (red), SimpleQA (blue), and MedQA (violet).

| Total budget constraint | Domain | Approach | Success Rate (avg) | No. tools used | Budget spent |
|---|---|---|---|---|---|
| **Baselines** **Budget: NA** | | | | | |
| | GAIA | Identity (all tools) | 0.28 | 122 | 398 |
| | SimpleQA | Identity (all tools) | **0.88** | 122 | 398 |
| | MedQA | Identity (all tools) | 0.84 | 122 | 398 |
| **Knapsack Composers** **Budget: $10** | | | | | |
| | GAIA | Offline knapsack | 0.19 | 3 | 9 |
| | | Online knapsack | 0.16 | 1 | 3 |
| | SimpleQA | Offline knapsack | 0.16 | 3 | 9 |
| | | Online knapsack | 0.82 | 1 | 8 |
| | MedQA | Offline knapsack | 0.82 | 3 | 9 |
| | | Online knapsack | 0.82 | 1 | 3 |
| **Knapsack Composers** **Budget: $30** | | | | | |
| | GAIA | Offline knapsack | **0.31** | 10 | 30 |
| | | Online knapsack | **0.31** | 2 | 11 |
| | SimpleQA | Offline knapsack | 0.80 | 10 | 30 |
| | | Online knapsack | 0.82 | 4 | 17 |
| | MedQA | Offline knapsack | **0.85** | 8 | 24 |
| | | Online knapsack | **0.85** | 1 | 3 |

Table 12: Robustness testing on SimpleQA with Claude 3.5 Sonnet across three runs. As shown, the standard deviations are low across all metrics, indicating that performance is stable and consistent across independent runs. Bolded row marks the highest success rate.

| Method | Budget | Success Rate | # Tools | Total Price | Avg Duration (s) | Tool Calls/Example |
|---|---|---|---|---|---|---|
| Identity | No limit | $0.840 \pm 0.033$ | 122 | $398.00 | $13.03 \pm 0.62$ | $2.55 \pm 0.04$ |
| Offline-KP | $10 | $0.213 \pm 0.009$ | 3 | $9.00 | $14.68 \pm 0.16$ | $3.95 \pm 0.01$ |
| Offline-KP | $30 | $0.207 \pm 0.009$ | 10 | $30.00 | $17.23 \pm 0.29$ | $4.25 \pm 0.03$ |
| Online-KP | $10 | $0.860 \pm 0.028$ | 3 | $9.00 | $12.44 \pm 1.04$ | $2.77 \pm 0.05$ |
| **Online-KP** | **$30** | $\mathbf{0.873 \pm 0.034}$ | **3** | **$14.00** | $\mathbf{11.71 \pm 0.68}$ | $\mathbf{2.81 \pm 0.01}$ |

