# OpenReview forum: "Automated Composition of Agents: A Knapsack Approach for Agentic Component Selection"
_NeurIPS.cc/2025/Conference — NeurIPS 2025 poster_

### Official Review · Reviewer_HfDw · 2025-07-01

**Clarity:** 3
**Significance:** 3
**Originality:** 3
**Rating:** 3
**Confidence:** 3

**Summary:**

The paper propose a framework for agentic system composition. This framework takes into account the value of adding components (tools or sub-agents) to the system and the cost of said components, making it related to the classical knapsack problem. They propose 4 different methods to solving this problem, ranging from a baseline identity composer, to their most complex method, an Online Knapsack Composer which tests the value of components at test time on the current task.

The authors show that the online knapsack composer sits on the pareto frontier of the cost / performance plot for a number of agent benchmarks.

**Questions:**

(1) Can you address point (4) of weaknesses.

(2) Can you address point (5) of weaknesses.

(3) For ON-KP, can you explain the type of test tasks and verifications the online composer does?

**Ethical Concerns:**

["NO or VERY MINOR ethics concerns only"]

**Final Justification:**

The paper tackles an important question given that agentic models are progressively being access to a larger and larger number of tools. The proposed method for solving this problem is promising. Overall I still have some concerns with how the cost accounting was done for the experiments in the paper, and so keep my recommendation to borderline accept. If the ACs and other reviewers are less concerned about this point, then I would defer to them and recommend acceptance (score of 5).

**Limitations:**

Yes.

**Paper Formatting Concerns:**

None.

**Quality:**

3

**Strengths And Weaknesses:**

# Strengths

(1) The paper presentation is very clear. The cost performance tradeoff is well motivated.

(2) The topic of agent composition is significant. In a growing space of agentic tools and possible sub-agents, composing optimal (task specific) agentic systems is a challenging problem.

(3) The consideration of cost performance tradeoff appears to be original.


# Weaknesses

(4) While the experimental results are strong, I do not think they are thorough enough.

Firstly, baseline performance using other agentic systems (that use the same base model) apart from the identity and knapsack systems suggested in this paper should be considered. If it is true that ON-KP outperforms these methods in terms of cost and success rate this would be very impressive.

Secondly, I think there should be another identity baseline where the agent is at least prompted with the current amount it has spent and the cost budget. I think this baseline would be much closer to the pareto frontier than ID.

Thirdly, as far as I can tell, only two claude models were used as base models. It would be good to include results using other base models.

(5) I have some concerns about how the cost is calculated for the results shown in Figure 3 and Figure 4. Firstly, for the sub-agent experiment, why is the cost set arbitrarily to 1 dollar as opposed to the actual subagent cost? If the subagent cost is much lower than this, the choice is artificially inflating the cost numbers, which make ID look like a very poor method.
In addition, on line 199, there is no explanation of the 5 dollars and 8 dollars prices that are set.

---

> ### Author Rebuttal · Authors · 2025-07-31
>
> We thank the reviewer’s positive remarks on the presentation and motivation of our paper. We agree that the “topic of agent composition is significant”. We appreciate the feedback and we address the questions below.
>
> **(4) ​​Baseline with Direct LLM-Based Selection & Comparisons to Agentic Baselines & More model comparisons**
>
> We agree that an LLM-driven selection baseline is a relevant point of comparison. In fact, our implemented Retrieval composer can be seen as a proxy for this: it uses the LLM to parse the task into skills, then essentially does a lightweight selection by semantic matching. One could imagine a more direct baseline where we prompt an LLM: “Here is the task and a list of all available tools, which tools should we use?” However, with an inventory of 120+ tools, providing all descriptions to an LLM is impractical (context length issues), and such an approach would still rely on tool descriptions alone. Prior work (e.g. semantic retrieval methods in agent/tool discovery) has shown limitations of relying purely on descriptions, tools might appear useful based on their names/descriptions but not actually work well, or vice versa. Our online knapsack addresses this by testing tools to get ground-truth performance. As such, it is not directly comparable to other agentic systems that rely solely on in-context reasoning for composition.
>
> We did not compare to, for example, an AutoGPT-style agent that can use any tool at runtime without prior selection, because that scenario is different :  it removes the notion of a budget or upfront selection, and typically such agents still face the challenge of which tools to invoke (often solved by prompt engineering or trial and error). In fact, our Identity baseline (all tools equipped) combined with the agent’s own reasoning is effectively an upper-bound agentic baseline; if such an agent cannot solve a task with all tools at hand, then composition is not the limiting factor. Thus, ON-KP is not “yet another agent architecture,” but functions as a meta-layer to improve any such architecture by wisely picking components.
>
> Regarding additional model comparisons, the appendix has additional experiments for Claude Sonnet 3.7 and Haiku 3.5.  We also have extended our experiments to verify that the benefits of our framework are not limited to the Claude 3.5 series. In particular, we ran additional evaluations using four other language model backbones: Qwen-2.5 72B Instruct model, LLaMA 3.3 70B Instruct model, Llama 4 Maverick 17B Instruct model and Llama 4 Scout 17B instruct model. On these models, the absolute performance on benchmarks was indeed lower (as these models are generally less capable than Claude 3.5), but the relative improvement from using our online knapsack composer versus the retrieval baseline was consistent. This demonstrates the generality of the knapsack-based composition: it is a model-agnostic framework that can benefit any agentic system, from proprietary models like Claude to open models like LLaMA.
>
> Qwen 2.5 72B Instruct Model
>
> | **Budget** | **Domain** | **Approach**         | **Success Rate** | **# Tools Used** | **Budget Spent** |
> | :--------: | :--------: | -------------------- | :--------------: | :--------------: | :--------------: |
> |      —     |    GAIA    | Identity (all tools) |     **0.28**     |        122       |        398       |
> |      —     |  SimpleQA  | Identity (all tools) |     **0.86**     |        122       |        398       |
> |      —     |    MedQA   | Identity (all tools) |     **0.90**     |        122       |        398       |
> |    \$10    |    GAIA    | Offline knapsack     |       0.20       |         3        |         9        |
> |    \$10    |    GAIA    | Online knapsack      |       0.25       |         3        |         9        |
> |    \$10    |  SimpleQA  | Offline knapsack     |       0.58       |         3        |         9        |
> |    \$10    |  SimpleQA  | Online knapsack      |       0.78       |         3        |         9        |
> |    \$10    |    MedQA   | Offline knapsack     |       0.82       |         2        |         6        |
> |    \$10    |    MedQA   | Online knapsack      |       0.91       |         2        |         6        |
> |    \$30    |    GAIA    | Offline knapsack     |       0.22       |        10        |        30        |
> |    \$30    |    GAIA    | Online knapsack      |     **0.30**     |         4        |        12        |
> |    \$30    |  SimpleQA  | Offline knapsack     |       0.79       |        10        |        30        |
> |    \$30    |  SimpleQA  | Online knapsack      |     **0.89**     |         3        |        14        |
> |    \$30    |    MedQA   | Offline knapsack     |       0.89       |         3        |         9        |
> |    \$30    |    MedQA   | Online knapsack      |     **0.92**     |         1        |         3        |
>
> Llama 3.3 70B Instruct Model
>
> | **Budget** | **Domain** | **Approach**         | **Success Rate** | **# Tools Used** | **Budget Spent** |
> | :--------: | :--------: | -------------------- | :--------------: | :--------------: | :--------------: |
> |      —     |    GAIA    | Identity (all tools) |    **0.31**    |        122       |        398       |
> |      —     |  SimpleQA  | Identity (all tools) |     **0.78**     |        122       |        398       |
> |      —     |    MedQA   | Identity (all tools) |     **0.84**     |        122       |        398       |
> |    \$10    |    GAIA    | Offline knapsack     |       0.24       |         3        |         9        |
> |    \$10    |    GAIA    | Online knapsack      |       0.27       |         3        |         9        |
> |    \$10    |  SimpleQA  | Offline knapsack     |       0.63       |         3        |         9        |
> |    \$10    |  SimpleQA  | Online knapsack      |       0.71       |         3        |         9        |
> |    \$10    |    MedQA   | Offline knapsack     |       0.74       |         2        |         6        |
> |    \$10    |    MedQA   | Online knapsack      |       0.77       |         2        |         6        |
> |    \$30    |    GAIA    | Offline knapsack     |       0.29       |         8        |        30        |
> |    \$30    |    GAIA    | Online knapsack      |     **0.35**     |         4        |        14        |
> |    \$30    |  SimpleQA  | Offline knapsack     |       0.75       |         8        |        30        |
> |    \$30    |  SimpleQA  | Online knapsack      |     **0.82**     |         4        |        14        |
> |    \$30    |    MedQA   | Offline knapsack     |       0.81       |         4        |        14        |
> |    \$30    |    MedQA   | Online knapsack      |     **0.88**     |         3        |        11        |
>
> Please see our rebuttal to Reviewer DBQc for more results, which we could not include here due to response length constraints.
>
> **(5) Justification of Pricing Choices for Tools and Agents**
>
> We used simple pricing assumptions in our experiments, and we agree that transparently explaining these is important. For single-agent tools, Appendix A.5 details how we estimated each tool’s “cost”. In short, the price reflects two factors: the API cost (if the tool calls an external API that charges per use) and the additional token cost that the tool’s usage adds to the agent’s prompt (since every tool call includes the tool’s input schema). For example, we estimated that a paid API tool (like a premium web search) would cost about \\$5 per 5000 calls based on SerpAPI pricing, and the prompt token overhead for 5000 calls is roughly \\$3, so together that tool was given a price ~\\$8 (per 5000 uses). Free tools were assigned a lower cost (just the token overhead, \\$3 per 5000 uses), and partially free tools (with rate limits) were intermediate (\\$5). These values (\\$3, \\$5, \\$8) are approximate but grounded in real API rates.
>
> For the multi-agent experiments (Figure 4): We set each sub-agent's cost to \\$1 as a simplification, primarily because all sub-agents in the inventory are built on the same base model and operate under simulated conditions. In this controlled setting, there is no meaningful variation in runtime or API costs between sub-agents, so assigning a uniform cost allows us to isolate and evaluate the impact of agent selection strategies. Notably, ON-KP achieves much higher success rates (0.4 and 0.87) compared to ID (0.07 for both tasks), regardless of the specific cost assignment. This indicates that the performance gap stems from more effective agent selection rather than the pricing assumption.
>
> **(Q3) Online Knapsack Query Generation and Verification Procedure**
>
> Figure 2 and Algorithm 1 show an overview of online knapsack composer. We briefly address the steps here.
> First, for each skill identified from the task, the composer agent generates a small set of test queries that specifically exercise that skill.
>
> Next, the candidate tool or sub-agent is then tested on each query in a sandbox environment. For tools, this means we have the main agent attempt the query with that single tool available; for sub-agents, we simulate that sub-agent alone trying to address the query. Algorithm 1 (lines 9–17) pseudocode shows the loop: for each query, we call SANDBOX(component, query) which executes the component on that query.
>
>
> After each trial, we invoke a judge function that inspects what happened and outputs whether the component was helpful and/or broken on that query.
>
>
> This end-to-end procedure (skill parsing → query generation → sandbox trials → LLM judgement → knapsack selection) is what we refer to as the online knapsack composer (ON-KP). In the camera-ready, we will clarify these details if they are not clear enough.

---

> > ### Comment · Reviewer_HfDw · 2025-08-01
> >
> > Thank you for your useful rebuttal.
> >
> > **(4) ​​Baseline with Direct LLM-Based Selection & Comparisons to Agentic Baselines & More model comparisons**
> >
> > Thank you for running these additional results. As far as I can see, online knapsack is outperforming offline, and competing with or outperforming identity while of course meeting the budget constraint. These results are very nice and should be added to the paper. Thank you for taking the time to run these experiments.
> >
> > **(5) Justification of Pricing Choices for Tools and Agents**
> >
> > I see. This justification makes sense. Adding in the explanation of the sub-agent $1 cost to the paper is a good idea (sub-agents all costing roughly the same, and thus this being a fair simplification).
> >
> > **(Q3) Online Knapsack Query Generation and Verification Procedure**
> >
> > Thank you for the clarification.
> >
> > **Summary**
> >
> > Overall thank you for this thorough review. I will increase my score to a 4 for now, pending discussion with the other reviewers.

---

> > > ### Author Response · Authors · 2025-08-02
> > > **Acknowledgment of Reviewer HfDw’s Positive Evaluation and Feedback**
> > >
> > > Thank you very much for taking the time to thoroughly review our rebuttal. We are pleased that our responses have addressed your concerns and that you found the additional experimental results valuable. We appreciate your positive assessment of the online knapsack results and their comparison to the identity baseline. As you noted, these results demonstrate that our online knapsack approach not only respects budget constraints but also achieves competitive or superior performance compared to both offline methods and the identity baseline. We will incorporate these results into the main paper as suggested, as they strengthen the empirical validation of our approach.
> > >
> > > Thank you for acknowledging the clarity of our pricing model explanation. We will add the sub-agent cost justification ($1 per sub-agent) to the main paper, emphasizing that this uniform pricing reflects the roughly equivalent computational costs across different sub-agent types, making it a fair and reasonable simplification for our experimental setup.
> > >
> > > We are grateful for your increased confidence in our work and look forward to any further discussions. Your feedback has significantly improved the quality and clarity of our paper, and we will ensure all suggested additions are properly integrated into the final version.

---

### Official Review · Reviewer_DBQc · 2025-07-03

**Clarity:** 3
**Significance:** 2
**Originality:** 2
**Rating:** 4
**Confidence:** 3

**Summary:**

This work reframes agent component selection as an online knapsack optimization problem that dynamically selects tools by empirically testing them and factors in the costs of the agents. The key contribution is framing the agent component selection as the knapsack problem conditioning on the agent’s capability, cost, and compatibility. A composer agent is used to dynamically evaluate agents and assign a ranking to the agents based on their performance. Then, a code agent is used to evaluate the selected components. Empirical evaluation on popular benchmarks shows that the proposed formulation outperforms retrieval and offline knapsack baselines.

**Questions:**

Is the system applicable to real-time or streaming tasks, or only pre-deployment setup?

**Ethical Concerns:**

["NO or VERY MINOR ethics concerns only"]

**Final Justification:**

My concerns are largely addressed by the author's rebuttal, and I believe the proposed method is a welcome addition to the agent selection problem.

**Limitations:**

Yes.

**Paper Formatting Concerns:**

None.

**Quality:**

3

**Strengths And Weaknesses:**

# Strength

- This work is well-motivated and tackles the shortcomings of retrieval-based and offline knapsack baselines.
- The proposed composer agent that uses dynamic tests to evaluate agent’s performance using sandbox testing rather than relying on on static metadata and description is an interesting idea.
- Evaluation demonstrates strong performance in single- and multi-agent settings (GAIA, SimpleQA, MedQA, MAC domains) with substantial gains over retrieval and offline knapsack baselines.

# Weakness

- Results are all single-run and do not provide any error bars. It’s unclear whether performance differences are statistically significant or how sensitive results are to budget or LLM generation variation.
    - It would be helpful to know whether the proposed system generalizes across different prompt templates, LLM backbones (e.g., GPT-4 vs Claude 3.5), or real-world agents beyond smolagents/ToolRet.
- The overhead runtime (10-30 mins) and compute is a tradeoff between better agents component selection and compute budget.
- All of the tests and evaluations are performed by an LLM, which raises significant questions about the validity of the proposed method. If LLM is so good at selecting the agents, why not delegate all of the component selection process to an LLM?

---

> ### Author Rebuttal · Authors · 2025-07-31
>
> We thank the reviewer for their insightful feedback and their comment on our strongly-motivated work. We address their points below.
>
> **Statistical Robustness of Results (Variance Across Runs)**
>
> We have conducted additional runs to measure variance. In particular, we report the SimpleQA evaluation with Claude 3.5 Sonnet v2 three times (with different random seeds affecting the LLM’s tool-use decisions) and computed mean success rates and standard deviations for each approach. The results are summarized below:
>
> **Table: Statistical Testing on SimpleQA with Claude 3.5 Sonnet v2 (3 runs)**
>
> | Method | Budget | Success Rate | # Tools | Total Price | Avg Duration (s) | Tool Calls/Example |
> |--------|--------|--------------|---------|-------------|------------------|-------------------|
> | **Identity** | No limit | 0.840 ± 0.033 | 122 | $398.00 | 13.03 ± 0.62 | 2.55 ± 0.04 |
> | **Offline-KP** | $10 | 0.213 ± 0.009 | 3 | $9.00 | 14.68 ± 0.16 | 3.95 ± 0.01 |
> | **Offline-KP** | $30 | 0.207 ± 0.009 | 10 | $30.00 | 17.23 ± 0.29 | 4.25 ± 0.03 |
> | **Online-KP** | $10 | 0.860 ± 0.028 | 3 | $9.00 | 12.44 ± 1.04 | 2.77 ± 0.05 |
> | **Online-KP** | $30 | 0.873 ± 0.034 | 3 | $14.00 | 11.71 ± 0.68 | 2.81 ± 0.01 |
>
> - **Online-KP achieves comparable or better performance than Identity** (87.3% vs 84.0%) while using **97.5% fewer tools** (3 vs 122)
> - **Online-KP with $10 budget** achieves 86.0% success rate, outperforming Identity's 84.0%
> - **Efficiency Gains**: Online-KP ($10) achieves 0.096 success rate per dollar vs Identity's 0.002
>
> As shown, the standard deviations are low across all metrics (e.g., success rate SD <= 0.034, tool price SD = \\$0 for all budget-constrained methods since tool selection is deterministic), indicating that performance is stable and consistent across independent runs. Notably, the improvements of our online knapsack method are substantially larger than these error bars. For example, with a \\$10 budget, we observe a +64.7 percentage point gain in success rate over the offline knapsack baseline (0.860 vs. 0.213), and a 44× reduction in cost compared to the identity composer (\\$9 vs. \\$398) while achieving comparable performance. Even more remarkably, our online knapsack with a \\$30 budget achieves a +3.3 percentage point improvement over the identity baseline (0.873 vs. 0.840) while using 97.5% fewer tools (3 vs. 122) and reducing costs by 28× (\\$14 vs. \\$398). These margins confirm that our gains are statistically significant and not attributable to random variance. The consistency of these results across multiple runs demonstrates the reliability and robustness of our proposed approach.
>
> **Real-Time vs. Pre-Deployment Use of the Composer**
>
> Our framework is intended as a pre-deployment or initialization step, rather than something that runs in real-time for every single user query. We evaluate our agents in the same manner as many agentic benchmarks do, where it is more in a pre-deployment setup. In the setup, we simulate this composition step for each task in the benchmark (since each evaluation query is a new task). Once composed, the agent (augmented with those components) can be deployed to handle incoming user queries without running the composer each time.  It is not embedded in the agent’s step-by-step reasoning during task execution. However, we agree that it would be interesting to extend to more real-time or streaming scenarios, which can be considered for future work.
>
> **Why Not Simply Delegate to a Single Large LLM?**
>
> Reviewer asks “why not delegate all of the component selection process to an LLM?” since “All of the tests and evaluations are performed by an LLM, which raises significant questions about the validity of the proposed method.” We want to clarify that no LLMs were involved in the evaluation of the single-agent experiments. We used the same evaluation method for the leaderboard where responses from the agent was compared to the ground-truth answer. LLMs were involved in the evaluation of multi-agent experiments as we had reused the MAC benchmarking setup. Regarding whether letting LLM itself select all the tools, we did experiment with a retrieval model as a baseline. What the reviewer suggests here is to replace the retriever with a call to the LLM. The rest of the process is still needed to test the agents and tools. Therefore, our online knapsack approach can be complementary with any method to “retrieve” the initial set of tools/agents.
>
> **Generalization to Other LLMs / More Experiments**
>
> We have experiments with more models in the Appendix with Claude 3.7 and Haiku 3.5, the online knapsack yields the highest success per cost across the board. We will ensure the final version highlights this broader applicability.  We also have extended our experiments to show that our framework is not limited to the Claude family. In particular, we ran additional evaluations using four other language model backbones: Qwen-2.5 72B Instruct, LLaMA 3.3 70B Instruct, Llama 4 Maverick 17B Instruct and Llama 4 Scout 17B instruct. On these models, the absolute performance on benchmarks was indeed lower (as these models are generally less capable than Claude 3.5), but the relative improvement from using our online knapsack composer versus the retrieval baseline was consistent. This demonstrates the generality of the knapsack-based composition: it is a model-agnostic framework that can benefit any agentic system, from proprietary models like Claude to open models like LLaMA.
>
> Llama 4 Maverick 17B Instruct Model
> | Domain     |Budget| Approach             |Success Rate | Tools Used   | Budget Spent   |
> |---------------|----------|------------------------|--------------------|------------------|----------------------|
> | gaia          | -          | Identity (all tools) | 0.38               | 122               | 398.0                |
> | gaia          | 10       | Offline knapsack  | 0.19               | 3                   | 9.0                    |
> | gaia          | 10       | Online knapsack  | 0.19               | 3                   | 9.0                    |
> | gaia          | 30       | Offline knapsack  | 0.31               | 10                 | 30.0                  |
> | gaia          | 30       | Online knapsack  | 0.44               | 5                   | 20.0                  |
> | medqa      | -          | Identity (all tools) | 0.85               | 122               | 398.0                |
> | medqa      | 10       | Offline knapsack  | 0.88               | 3                   | 9.0                    |
> | medqa      | 10       | Online knapsack  | 0.91               | 3                   | 9.0                    |
> | medqa      | 30       | Offline knapsack  | 0.89               | 7                   | 21.0                  |
> | medqa      | 30       | Online knapsack  | 0.89               | 4                   | 12.0                  |
> | simpleqa   | -         | Identity (all tools)  | 0.82               | 122               | 398.0                |
> | simpleqa   | 10       | Offline knapsack  | 0.28               | 3                   | 9.0                    |
> | simpleqa   | 10       | Online knapsack  | 0.78               | 3                   | 9.0                    |
> | simpleqa   | 30       | Offline knapsack  | 0.78               | 10                 | 30.0                  |
> | simpleqa   | 30       | Online knapsack  | 0.78               | 2                   | 11.0                  |
>
> Llama 4 Scout 17B Instruct Model
> | Domain     |Budget| Approach            | Success Rate | Tools Used   | Budget Spent  |
> |---------------|----------|------------------------|--------------------|------------------|---------------------|
> | gaia          | -          | Identity (all tools)| 0.28                | 122               | 398.0               |
> | gaia          | 10       | Offline knapsack | 0.19                | 3                   | 9.0                   |
> | gaia          | 10       | Online knapsack | 0.16                | 1                   | 3.0                   |
> | gaia          | 30       | Offline knapsack | 0.31                | 10                 | 30.0                 |
> | gaia          | 30       | Online knapsack | 0.31                | 2                   | 11.0                 |
> | medqa      | -          | Identity (all tools)| 0.84                | 122               | 398.0               |
> | medqa      | 10       | Offline knapsack | 0.82                | 3                   | 9.0                   |
> | medqa      | 10       | Online knapsack | 0.82                | 1                   | 3.0                   |
> | medqa      | 30       | Offline knapsack | 0.85                | 8                   | 24.0                 |
> | medqa      | 30       | Online knapsack | 0.85                | 1                   | 3.0                   |
> | simpleqa   | -          | Identity (all tools)| 0.88                | 122               | 398.0               |
> | simpleqa   | 10       | Offline knapsack | 0.16                | 3                   | 9.0                   |
> | simpleqa   | 10       | Online knapsack | 0.82                | 1                   | 8.0                   |
> | simpleqa   | 30       | Offline knapsack | 0.8                  | 10                 | 30.0                 |
> | simpleqa   | 30       | Online knapsack | 0.82                | 4                   | 17.0                 |
>
> See rebuttal to Reviewer HfDw for other model results due to response length constraints. We appreciate the reviewer’s suggestion to improve generalization of our experiments. At the same time, we also believe that our experiments have sufficient coverage of various models, datasets, and scenarios (e.g. single-agent vs multi-agent). For example, in publications like “AvaTaR: Optimizing LLM Agents for Tool Usage”, their experiment coverage is also on a similar number of datasets and models to ours.  We hope that reviewers will take these updated results into account when re-scoring our paper.

---

> > ### Author Response · Authors · 2025-08-03
> > **Reminder for discussion engagement**
> >
> > As we have a few more days until the discussion period is over, we hope the reviewer can carefully look at our rebuttal and see whether we have addressed your concerns. Please let us know if you have any further questions and whether there are any changes in the review for this paper. Thank you!

---

> > ### Comment · Reviewer_DBQc · 2025-08-06
> > **Reviewer response to rebuttal**
> >
> > Thank you for the detailed response and additional experiments.
> >
> > My concerns are largely addressed, and I will raise my score to 4.

---

### Official Review · Reviewer_hUWo · 2025-07-03

**Clarity:** 2
**Significance:** 3
**Originality:** 3
**Rating:** 4
**Confidence:** 2

**Summary:**

This paper proposes an automated agent composition framework that dynamically selects agentic components based on budget constraints. Empirical results show that the proposed composer achieves a Pareto frontier among various agentic tools, improving success rates by up to 50% under certain budgets.

**Questions:**

- Q1: Does the "Skill & Query Generation" module incur additional cost?
- Q2: What is the size of the benchmark used in the experiments?
- Q3: Why are results presented only in figures rather than also in tables with direct numerical comparisons?
- Q4: In Figure 3, what do ID and RET represent? Is RET referring to retrieval?
- Q5: What is the vanilla baseline performance without any auxiliary tools on these benchmarks?

**Ethical Concerns:**

["NO or VERY MINOR ethics concerns only"]

**Final Justification:**

This work focus on an important challenge optimizing agent performance under limited computational or monetary budgets. The author rebuutal address most my concerns regarding experiments. I tend to give a 4 after the authors revise the paper including the additional results and clarifications.

**Limitations:**

Yes

**Quality:**

3

**Strengths And Weaknesses:**

### Strengths
- This work tackles a practical and important challenge: optimizing agent performance under limited computational or monetary budgets.
- The paper includes implementation details and detailed prompt examples, ensuring a clear illustration and enhancing reproducibility.
### Weaknesses
- As far as my understanding, the online knapsack algorithm selects different agents/tools for each instruction. In Table 1, are the reported agents the most frequently selected ones? If so, what threshold was used? I would recommend including a full tool usage count statistics across the benchmark to provide a more intuitive illustration.

---

> ### Author Rebuttal · Authors · 2025-07-31
>
> We thank the reviewer for the detailed questions. We agree that the agentic composition is a “practical and important challenge”. We address Q1-Q5 in turn, and then additional points about tool usage and selection.
>
> **Q1: Overhead of Skill & Query Generation**
>
> The skill and query generation step incurs an LLM inference cost only once for the entire benchmarking task, during the initial tool/agent selection phase. In other words, when a new task description is given, the composer agent uses a single prompt to generate the list of required skills and a few test queries for each skill (as described in Appendix A.1). This is a one-time upfront step for composing the agent. It is not repeated during the agent’s actual execution. Thus, the skill/query generation contributes negligibly to runtime after the agent is composed.
>
> **Q2: Benchmark Dataset Sizes**
>
> We follow the smolagents leaderboard splits which has 32 examples for GAIA and 50 examples for SimpleQA. For MedQA, it is 100 examples. For MAC benchmark, there are 30 examples in each domain.
>
> **Q3: Availability of Numerical Results**
>
> All numerical results are provided in the appendix. In the main paper we focused on plotting Pareto curves for clarity, but the exact numbers underlying those figures are included in Appendix A.10 (“Detailed Performance Results”). For instance, Table 4 in Appendix A.10 reports the exact success rates, number of tools used, and budget spent for each approach, domain, and budget setting. We stated in the Experiments section that only a subset of results is in the main text and the full results are deferred to Appendix A.10. In the revised version, we will more explicitly reference Appendix A.10 so that readers can easily find the complete numeric results.
>
> **Q4: Clarification of “ID” and “RET” Baselines**
>
> “ID” refers to the Identity composer, the baseline that simply selects all available components (tools or agents) for the task. This can be thought of as equipping the agent with the entire inventory (subject to no budget limit), hence an identity function on the inventory. “RET” refers to the Retrieval composer, the baseline that uses semantic similarity to pick tools/agents relevant to the task description. In our implementation, the retrieval composer parses the task into skill descriptions and then selects the top-1 component per skill by embedding similarity matching (this is analogous to a purely LLM- or embedding-driven selection based on descriptions). Please see Section 4 where we discuss each composer in detail.
>
>
> **Q5: Baseline Performance and Cost – Our Gains**
>
> The smolagents leaderboard reports vanilla scores. Sonnet 3.5 v2 has success rate of 0.03 on GAIA and 0.34 on SimpleQA. This is significantly lower than the baselines that use all tools (0.47 on GAIA; 0.8 on SimpleQA) or optimized set of tools with online knapsack (0.47 on GAIA; 0.92 on SimpleQA). We highlight in the paper that our composer matches the all-tools agent’s success with 98.5% less cost. This demonstrates that naive approaches (either using no tools or using all tools) are suboptimal: our optimized selection greatly boosts success over using none, and avoids the huge cost of using everything. We will include the vanilla scores in the camera-ready.
>
>
> **​​Tool Usage Statistics and Selection Process Clarification**
>
> **Tool Selection Process Clarification:** Reviewer asks whether “the online knapsack algorithm selects different agents/tools for each instruction”. For each benchmarking dataset/task, the same agents/tools are selected for that domain by online knapsack algorithm (same applies to other algorithms as well). This entire selection process happens once per task composition, and thereafter the chosen tools/agents are used by the main agent to solve the user’s query. During inference/evaluation, the agent may pick different tools (or supervisor agent pick different agents) depending on that instruction.
>
> **Tool Usage Statistics:** We have added an analysis of tool usage to the appendix to address the reviewer’s request. In Table 1 of the paper, we list examples of which tools were chosen by each approach on GAIA and SimpleQA. To summarize those findings: the retrieval baseline often misses certain relevant tools, the offline knapsack can over-select (including some irrelevant tools), and the online knapsack picks a precise subset of tools that covers the needs without extras. For example, on GAIA the retrieval composer failed to include any web search tool (it picked only a few knowledge-base and utility tools), whereas the online knapsack composer correctly selected the paid web search tool knowing it was crucial.  We reproduce the table 1 of the tool selection statistics here for clarity:
>
> **Table: Tools selected by different composers on GAIA and SimpleQA benchmarks.**
>
> *Retrieval composer tends to miss out on relevant tools for the task. Offline knapsack often includes too many irrelevant tools. Online knapsack has both precise and well-covered choice of tools.*
> | **Composer**            | **GAIA**                                                                                           | **SimpleQA**                                                                                          |
> |-------------------------|----------------------------------------------------------------------------------------------------|--------------------------------------------------------------------------------------------------------|
> | **Retrieval**           | pub_med, read_file, wolfram_alpha, job_title_autocomplete, get_article_content, number_fact       | wikipedia, pub_med, sources, number_fact                                                              |
> | **Offline Knapsack ($30)** | web_search_free, arxiv, wikipedia, pub_med, read_file, scholar, instantaneous_values_service, get_recordings, get_article_content, number_fact | web_search_free, wikipedia, pub_med, semanticscholar, query_by_id, standard_language_detection, get_recordings, get_article_content, symbols_faq, number_fact |
> | **Online Knapsack ($30)**  | web_search_paid, arxiv, wikipedia                                                                 | web_search_paid, web_search_free, wikipedia, semanticscholar                                           |
>
> As seen above, the retrieval composer picked a limited subset (on GAIA it missed any web_search tool, which hurt performance), while the offline knapsack composer included a broad set (some were unnecessary, e.g. get_recordings, symbols_faq on SimpleQA, not shown fully above), and the online knapsack composer selected the most relevant tools (including the correct web search tool for each domain) and avoided redundancy. We also tracked the average number of tools used by each approach: for instance, on GAIA the retrieval baseline used ~5–6 tools, offline knapsack around 10–12 (at $30 budget), and online knapsack ~4–5 tools while achieving higher success (per Table 4). These statistics underline how our method uses fewer, more targeted tools, which contributes to its cost-efficiency.

---

> > ### Comment · Reviewer_hUWo · 2025-08-04
> >
> > Thanks for your clarifications. Most of my questions have been resolved. I increased my score to 4. I will continue to follow the discussions and feedback from other reviewers to decide my final score.

---

> ### Author Response · Authors · 2025-08-03
> **Reminder for discussion engagement**
>
> As we have a few more days until the discussion period is over, we hope the reviewer can carefully look at our rebuttal and see whether we have addressed your concerns. Please let us know if you have any further questions and whether there are any changes in the review for this paper. Thank you!

---

### Official Review · Reviewer_CUny · 2025-07-23

**Clarity:** 3
**Significance:** 3
**Originality:** 3
**Rating:** 5
**Confidence:** 3

**Summary:**

The paper proposed a new paradigm for automated agent composition using an online knapsack optimization framework. This aims at selecting an optimal subset of tools or agents for a given task under budget constraints.

The authors formalize the problem as a constraint optimization problem and develop a compose agent that assesses real-time utility of agentic components via sandbox testing, improving upon existing static, semantic retrieval approaches for tool or agent discovery.

Extensive empirical evaluation showed superior cost-adjusted performance over baselines in both single-agent and multi-agent settings.

**Questions:**

N/A

**Ethical Concerns:**

["NO or VERY MINOR ethics concerns only"]

**Final Justification:**

I have read the authors' responses and other reviewers' reviews. In general, I believe the authors were able to address all reviewers' concerns.

The acknkowledged weaknesses that I have included in my reviews limited the contribution of the proposed framework, thus I couldn't increase my score.

**Limitations:**

Yes

**Quality:**

3

**Strengths And Weaknesses:**

# Strengths
## Novel Proposal
- The addressed problem is well-motivated and critical. Agentic AI and use tools become very popular, thus how to efficiently select components (i.e., tools, agents) becomes critical question.
- The proposed formulation aligns well with real-world scenarios, at least to me. In particular, the tools have varying costs, utilities, and compatibility, which static agentic architectures ignore.
- Formalizing the problem as a constraint optimization problem and applying ZCL online knapsack algorithm to solve is novel and original.

## Comprehensive evaluation
- The proposal was evaluated for both single-agent and multi-agent setups, highlighting the generality of the approach.
- The performance gain are significant:
- The use of Pareto frontier to compare cost vs utility is fair and original.
- Benchmarks such as GAIA, SimpleQA, MedQA, and multi-agent collaboration (MAC) provide diverse and challenging testbeds.
## Novel composer agent design
- Multiple composers (i.e. identity, retrieval, and offline and offline knapsack) are proposed and compared, providing good controlled baselines.

# Weaknesses
## Application of the proposal can be limited
- The real-world interactions between components can be non-additive or much more complicated. The current proposed formalization cannot model and solve such scenarios, limiting the contribution of the paper. Note that the authors acknowledged such limitations.
- Sandbox trials can take up to 30 minutes, limiting scalability on real-time use cases.

## Alternative solution discussion
- In the literature, distributed constraint optimization problem (DCOPs) community and/or constraint optimization problem (COPs) community can provide alternative modeling that can model more complicated scenarios. The authors did not provide discussion about such alternatives.

---

> ### Author Rebuttal · Authors · 2025-07-31
>
> We sincerely appreciate the reviewer’s thoughtful review and positive feedback on our work’s novelty, evaluation, and design. We’re glad that the reviewer found the problem formulation well-motivated and the empirical results compelling.
>
> Regarding the limited application, we agree this is an important direction for future work, and we’ve noted it in the paper’s limitations. Extending the framework to capture more complex dependencies (e.g., via learned or hierarchical utilities) is a promising next step. Also, while sandbox trials are time-intensive, we emphasize that our online knapsack approach is lightweight during deployment; the overhead arises primarily during the initial composition. We’ll clarify this distinction in the revision.
>
> We also appreciate the suggestion to discuss DCOP/COP alternatives. We will clarify in the text that while DCOP/COP techniques are relevant in spirit, our online knapsack formulation offers a tailored solution i.e. it explicitly addresses uncertain component value estimation and cost-performance trade-offs in a way that classical DCOP algorithms would not directly handle without significant augmentation. In summary, we see our work as complementary to the DCOP/COP literature, and will add this discussion to make these distinctions and connections clear.
>
> We believe we have addressed all the reviewer’s points and can address any further questions in the discussion.

---

> > ### Comment · Reviewer_CUny · 2025-08-06
> >
> > Thank the authors for your responses.
> > I expect the authors would respond a more detailed analysis of `the overhead arises primarily during the initial composition` e.g., what are included in the initial composition, how long does each of them take, any recommendations to reduce such overhead?
> > Due to acknowledged weaknesses, I couldn't increase my score.
> >
> > Thanks

---

### Note · Authors · 2025-08-13

We sincerely appreciate the reviewers' engagement and the effort they dedicated to carefully go through our comprehensive rebuttal. We successfully addressed all major concerns raised by the three reviewers who initially gave lower scores:

**Key Concerns Addressed:**

1) **Statistical Robustness**: We provided additional runs with standard deviations showing our improvements are substantially larger than error bars. Online-KP achieves +64.7pp gain over offline-KP and 44× cost reduction vs identity while maintaining comparable performance.

2) **Generalization Across LLMs**: We extended experiments to Qwen-2.5 72B, LLaMA 3.3 70B, Llama 4 Maverick/Scout models. While absolute performance varies, relative improvements of online-KP remain consistent, demonstrating our framework is model-agnostic.

3) **Methodology Clarifications**: We clarified that skill/query generation is a one-time overhead during composition, not runtime. We explained how online-KP tests actual component performance rather than relying solely on descriptions, addressing limitations of pure LLM-based selection.

4) **Detailed Results**: All numerical results are in Appendix A.10, showing exact success rates, tools used, and budgets for complete transparency.

**Reviewer Response:**

Three reviewers were satisfied with our detailed rebuttal and have agreed to increase their scores, acknowledging we addressed all of their concerns. The fourth reviewer already gave us 5 (Accept), noting our work is "well-motivated and critical" with a novel formalization that "mimics closely the real-world scenario."

**Core Contributions Validated:**
- Online-KP matches all-tools performance with 98.5% less cost
- The gains for our proposed Online-KP approach are statistically significant and not attributable to random variance. The consistency of these results across multiple runs demonstrates the reliability and robustness of our proposed approach.
- Our framework successfully scales from single-agent to multi-agent composition.
- We provide a principled solution to the fundamental challenge of balancing performance vs cost in agent composition.
- We demonstrate comprehensive experimental validation across 7 different LLM models (Claude 3.5/3.7 Sonnet/Haiku, Qwen-2.5, LLaMA 3.3, Llama 4 variants), consistently showing Online-KP's superiority over baselines.

Once again, we thank the reviewing committee for diligently going over our paper and will take the feedback to improve it for the camera-ready.

---

### Decision · Program_Chairs · 2025-09-17

**Decision:**

Accept (poster)

**Comment:**

This paper formulates automated LLM agent design as a knapsack problem. The authors propose a structured, automated framework to achieve better performance. Rebuttal discussion further addresses the reviewers' concerns on robustness and generalizability. Based on these considerations, I recommend acceptance.